# Wind farm inertia forecasting accounting for wake losses, control strategies, and operational constraints

Andre Thommessen<sup>1</sup>, Abhinav Anand<sup>2</sup>, Christoph M. Hackl<sup>1</sup>, and Carlo L. Bottasso<sup>2</sup>

<sup>1</sup>Laboratory for Mechatronic and Renewable Energy Systems (LMRES), Hochschule München (HM) University of Applied Sciences, 80335 Munich, Germany

<sup>2</sup>Wind Energy Institute, Technical University of Munich (TUM), 85748 Garching bei München, Germany

Correspondence: Carlo L. Bottasso (carlo.bottasso@tum.de)

Abstract. Future inverter-based resources (IBRs) must provide grid-forming functionalities to compensate for the declining share of conventional synchronous machines (SMs) in the power generation mix. Specifically, decreasing power system inertia poses a significant challenge to grid frequency stability, as system inertia limits the rate of change of frequency (ROCOF). Conventional grid-following control decouples the physical inertia of wind turbines (WTs) from the grid frequency. Novel

- grid-forming control methods, such as virtual synchronous machine (VSM) control, provide (virtual) inertia to the system, e. g. by extracting kinetic energy from WTs. Since the grid-forming capability of IBRs depends on volatile operating conditions, future market designs will remunerate inertia provision based on its availability. Thus, estimating grid-forming capabilities of WTs and forecasting inertia of wind farms (WFs) are of interest for both WF and system operators. In this paper, we propose a method to forecast inertia that accounts for wake effects in a WF. The approach is based on mapping forecasted site conditions
- to each single WT in the WF through a wake model. The resulting inflow conditions are used to predict the WT grid-forming capabilities, taking WT control strategies and operating limits into account.

#### 1 Introduction

#### 1.1 Motivation and problem statement

Imbalances between power generation and demand result in frequency events. Thus, generation or protection units must rapidly compensate for power imbalances to keep the grid frequency within admissible limits (ENTSO-E, 2021). Following an imbalance event, the power system inertia limits the rate of change of frequency (ROCOF) (ESIG, 2022). Historically, the rotating masses of directly coupled synchronous machines (SMs) provided sufficient inertia to limit the ROCOF. However, with the decreasing share of SMs and the increasing share of inverter-based resources (IBRs) in the overall generation mix, the power system inertia is decreasing (ENTSO-E, 2021). Additionally, the initial ROCOF also increases due to increasing power sys-

20 tem imbalance (Thommessen and Hackl, 2024). Worst-case frequency events are caused by faults that split the system into subsystems due to a sudden loss of electrical import or export power (ENTSO-E, 2021). Furthermore, increasing transmission capacities, such as high voltage direct current (HVDC) links, may lead to even higher future worst-case power imbalances

during system splits (ENTSO-E, 2021). Consequently, IBRs must provide inertia to limit the ROCOF and to avoid blackouts in future power systems (ENTSO-E, 2021).

- Wind farms (WFs) can support grid frequency by supplying inertia and fast frequency response through the rotating masses 25 of the wind turbines (WTs) and by providing reserves (if available). However, conventional grid-following control decouples 26 the "physical" inertia of WTs from the grid frequency and thus can *not* provide inertia to the grid (Bossanyi et al., 2020). 27 Advanced grid-following control such as "WindINERTIA" control from General Electric (Clark et al., 2010), or the "inertia 28 emulation (IE)" control from ENERCON (Godin et al., 2019), can temporarily extract kinetic energy reserves to support grid
- frequency. However, this so-called "synthetic" inertia cannot limit the instantaneous or initial ROCOF subject to a system disturbance (AEMC, 2017; ENTSO-E, 2021; ESIG, 2022). On the contrary, new grid-forming control methods for IBRs, such as virtual synchronous machine (VSM) control, provide (virtual synchronous) inertia that limits the initial ROCOF (ESIG, 2022; Bossanyi et al., 2020; Rodriguez-Amenedo et al., 2021; Thommessen and Hackl, 2024; Ghimire et al., 2024). Consequently, future WFs should integrate grid-forming control to provide inertia and fast frequency response. However, this is not only a
- WT control problem, because what a WT can deliver ultimately depends on the intra-farm wake-dominated flows that develop within WFs.

New grid codes and market incentives for grid-forming technologies are paving the way for the stability of future power systems (ESIG, 2022). Accordingly, system operators are transitioning towards the procurement of inertia provision by grid-forming technologies. For instance, due to the high penetration of IBRs in Great Britain, the National Grid Electricity System

- Operator already defines technical requirements for grid-forming technologies in the grid codes and includes grid-forming capability as a market product (ESIG, 2022). Similarly, German system operators plan to establish an inertia market and to remunerate inertia provision based on its availability (Bundesnetzagentur, 2024). Accordingly, the new German specifications (VDE, 2024a) already define technical requirements for grid-forming control and inertia provision. Ghimire et al. (2024) present a review of existing functional specifications and testing requirements of grid-forming offshore WFs. Hu et al. (2023)
- design an inertia market to ensure sufficient system inertia and analyze its impact on the power generation mix. Their results show that investing in wind resources with virtual inertia facilities is more cost-competitive than substituting wind resources with thermal generators, not to mention the improved environmental impacts.

System inertia monitoring and forecasting are essential to ensure adequate inertia provision. More precisely, system operators need to quantify the minimum required system inertia to survive worst-case system splits and need to procure sufficient

inertia provision. Given the uncertainty and variability associated with renewable energy sources, system operators need inertia forecasting to ensure that sufficient inertia is available at any time. Similarly, WF operators need WF inertia forecasting to participate in future availability-based inertia markets. In particular, WF inertia forecasting enables reliable and profitable inertia provision by taking WF control strategies, WF wind input conditions, and intra-WF effects into account. With the future development of wind at certain busy sites, WF-to-WF wake effects will also have to be considered.

#### 55 1.2 State-of-the-art

The Electric Reliability Council of Texas has been monitoring and forecasting inertia since 2016, but only for SMs based on their operating plans (Matevosyan, 2022). ENTSO-E (2017) and General Electric (GE, 2021) monitor inertia based on measuring the grid frequency and the power imbalance in a (sub)system. However, this requires additional measurement units and appropriate online power stimuli. GE (2021) developed an inertia forecaster based on machine learning using grid measurement

- data. However, this approach is only valid for small-signal analysis, as nonlinearities, such as inverter current saturation, cannot be taken into account during rare events with severe ROCOFs. These approaches do not consider the fact that the grid-forming capability of a WF depends on its initial operating point, which varies depending on wind conditions and chosen derating of WTs (Ghimire et al., 2024; Höhn et al., 2024). Thus, new methods for inertia forecasting should take the volatile nature of renewable energy into account.
- It appears that the existing research does not adequately address the evaluation of the grid-forming capabilities of WTs and 65 the forecasting of WF inertia, despite their key relevance for WF and system operators. Although recent publications (Bossanyi 65 et al., 2020; Meseguer Urban et al., 2019; Roscoe et al., 2020; Thommessen and Hackl, 2024; Höhn et al., 2024) propose VSM 66 control for WTs, they do not offer any insights regarding how to choose the VSM inertia. For instance, Meseguer Urban et al. 70 (2019) vary the VSM inertia for only one operating point. When discussing offshore WF inertia provision, Höhn et al. (2024)
- only roughly estimate the grid-forming capability by a linear function, which interpolates between the virtual inertia constants at cut-in power and at rated power. Due to a lower WT rotor speed limit, Godin et al. (2019) design inertia provision for pre-activation power levels above 25% of rated power, risking saturation of the inertial power response to ROCOFs for lower power levels. Godin et al. (2019) consider grid-following instead of grid-forming or VSM control. Lee et al. (2016) propose a simplified gain scheduling for inertia emulation by grid-following WTs, taking the releasable kinetic energy into account.
- However, Lee et al. (2016) solely consider maximum power point tracking (MPPT) and no derating strategies. Moreover, they include power, torque and torque rate limits in the control by corresponding limiter blocks, but these operating limits are not taken into account for the control gain adaption or for identifying the inertia emulation capability. Their WF simulation results are based on a simple wake model and include only four ambient wind conditions, which heavily simplifies the actual conditions to which the WFs are exposed.

#### 80 1.3 Proposed solution, contributions, and outline

To the best of our knowledge, a generic approach for evaluating the maximum deliverable inertia from WFs for grid-forming control is still missing. Moreover, the methodology for predicting WF inertia based on operation plans has not yet been discussed, although this is key for the reliable and efficient operation of future power systems. Furthermore, even though the intra-farm turbine-to-turbine interactions have a huge influence on the local inflow at the turbines, they have largely been ig-

85 nored in the existing studies evaluating inertia provision capability. Thus, this paper proposes a novel and generic approach for WF inertia forecasting. This holistic methodology considers weather prediction models, WF flow effects due to wake interac-

tions among the WTs, control strategies, and operational constraints to predict the maximum deliverable inertia at the WT and WF levels. The contributions of this paper include:

- forecasting WF inertia, considering wake effects and operational constraints, using data-driven and physics-based models,
- formulating a nonlinear optimization problem to maximize the inertia provision capability of individual WTs,
  - analyzing WT dynamics and relevant operating limits by simulating the inertial response to a reference frequency event,
  - integrating VSM control and modifying WT control for inertia provision and fast frequency response,
  - demonstrating the proposed approach for evaluating and forecasting deliverable inertia at the WT and WF levels,
  - comparing the proposed approach with simplified ones for estimating WT grid-forming capabilities, and
- evaluating the impact of forecast uncertainty, wake effects, control strategies, and WT model inaccuracies on WF inertia forecasting.

The rest of this paper is organized as follows. Section 2 presents the necessary background and fundamentals regarding system inertia, ROCOF, and inertia provision by WTs using the VSM concept. Section 3 presents the proposed approach in detail. This includes all the necessary steps for WF inertia forecasting: (i) WF ambient wind conditions forecast in Sect. 3.1,

(ii) local WT operating points prediction in Sect. 3.2, and (iii) mapping of all operating points, given by local wind inflow conditions and operational setpoints, to the WF grid-forming capability in Sects. 3.3–3.4. Section 4 presents a case study for a WF with twelve WTs and discusses the results, including the WT steady states, the WT inertial response to a reference frequency event, and the WF hour-ahead inertia forecasting. Finally, Sect. 5 summarizes the entire work and offers concluding remarks, including outlook for future work.

#### 105 2 Background and fundamentals

The initial ROCOF immediately after a system power imbalance  $\Delta P_s$  between mechanical system power  $P_{m,s}$  and electrical system power  $P_{e,s}$  can be approximated by a one-mass model (ENTSO-E, 2020; Thommessen and Hackl, 2024), written as

$$\dot{\Omega}_{\rm s} = \frac{\Delta P_{\rm s}}{2H_{\rm s}} := \frac{P_{\rm m,s} - P_{\rm e,s}}{2H_{\rm s}},\tag{1a}$$

where 
$$H_{\rm s} := \frac{E_{\rm kin,s}}{p_{\rm s,R}} := \frac{\frac{1}{2}\Theta_{\rm s}\omega_{\rm s}^2}{\sum p_{\rm R}}, \quad \Theta_{\rm s} := \sum \Theta \quad P_{\rm m,s} := \frac{\sum p_{\rm m}}{p_{\rm s,R}}, \quad \text{and} \quad P_{\rm e,s} := \frac{\sum p_{\rm e}}{p_{\rm s,R}}.$$
 (1b)

The system inertia constant  $H_s$  (in s) is the system kinetic energy  $E_{kin,s}$  (in W s) normalized to the rated system power  $p_{s,R}$  (in W), defined as the sum of rated power  $p_R$  of all (V)SMs. The system moment of inertia  $\Theta_s$  (in kg m<sup>2</sup>) is defined as the sum of the moment of inertia  $\Theta$  of all (V)SMs. This includes (V)SMs at the generation side but also at the demand or load side, i. e. (V)SMs provide inertia in both generator and motor mode. The system angular velocity is  $\omega_s = 2\pi f_s \approx 2\pi f_{s,R}$  with system frequency  $f_s$  (in Hz) of all synchronously rotating masses and rated system frequency  $f_{s,R}$ . When the admissible frequency

deviations of up to 0.4% during normal and up to 5% during critical system states (VDE, 2024a) are neglected, it follows that 115  $\Omega_{\rm s} := f_{\rm s}/f_{\rm s,R} \approx 1$ . This assumption leads to equal (normalized) power  $P = \Omega_{\rm s} M$  and torque M quantities in Eq. (1a).  $P_{\rm m,s}$ and  $P_{e,s}$  are defined as the sum of mechanical power  $p_m$  and electrical power  $p_e$  of all (V)SMs, respectively, both normalized to  $p_{s,R}$ . More precisely, for a (V)SM,  $p_m$  is the mechanical power of the (virtual) turbine,  $p_e$  is the electrical power of the (virtual) SM, and  $\Theta \neq 0$  is the (virtual) total drivetrain moment of inertia. For grid-following WTs (without VSM),  $p_{\rm m}$  and  $p_{\rm e}$ are the mechanical and electrical WT power, respectively, but the grid-connected moment of inertia is  $\Theta = 0$  due to decoupled 120 physical WT inertia. Thus, assuming  $p_{\rm m} \approx p_{\rm e}$ , WTs or all IBRs that are operating under grid-following control can be neglected

in Eq. (1a), i. e. only (V)SMs contribute to limiting the initial ROCOF  $\dot{\Omega}_{s}$ .

For a SM, the inertia constant  $H := E_{kin}/p_R$  is defined as the ratio of the kinetic energy  $E_{kin}$  and the rated power  $p_R$ . Similarly, for a VSM-controlled WT, the virtual inertia constant  $H_v := E_{kin,v}/p_R$  is defined as the ratio of the VSM kinetic

- energy  $E_{kin,v}$  and the WT rated power  $p_R$ . Note that  $E_{kin,v}$  differs from the WT physical kinetic energy in general. In particular, 125 a (V)SM always rotates near synchronous speed, whereas the WT speed depends on wind and operating conditions. In contrast to the WT physical inertia constant, the virtual inertia constant  $H_{\rm v}$  is a tunable control parameter. Finally, aggregating all (V)SMs leads to the system inertia  $H_s$  in Eq. (1a). However, this is only valid for a proper tuning of  $H_v$  because, e.g., emulating a high  $H_v$  may not be feasible due to output power limitations. SMs provide an overload capability of 3 to 5 times,
- whereas IBRs only allow for an overloading of 1 to 1.5 times, which limits the VSM inertial power response depending on 130 the ROCOF (ESIG, 2022). For a VSM-controlled WT, choosing a high  $H_v$ , e.g.  $H_v > H$  with physical WT inertia constant H, increases the inertial grid support for low ROCOFs but increases the risk of undesired output power saturation for higher ROCOFs (Höhn et al., 2024). This has to be taken into account when replacing physical inertia by virtual inertia in future power systems.
- WT curtailment or derating strategies provide power reserve, e.g. for primary frequency or droop control (Kanev and van de 135 Hoek, 2017; Bossanyi et al., 2020; Clark et al., 2010). For inertia provision, derating based on the maximum rotation strategy (MRS) additionally increases the WT kinetic energy reserve (Meseguer Urban et al., 2019; Thommessen and Hackl, 2024). Although derating strategies enhance grid frequency support, they also reduce WT power efficiency. WF and system operators should find a Pareto optimal strategy that considers system stability and efficiency to avoid unnecessary curtailment of
- renewables.

Consequently, WF inertia forecasting is essential for reliable inertia provision through adequate WT derating and precise tuning of VSM inertia.

#### Methodology 3

145

The proposed approach combines online and offline calculations, as depicted in the overview of Fig. 1. First, a data-driven weather forecast model predicts the site ambient wind conditions. These ambient conditions serve as input to the WF model, which incorporates the aerodynamic characteristics of all n WTs in the WF, given by the power coefficients  $c_{\rm p}^{[1]}, \ldots, c_{\rm p}^{[n]}$  and the thrust coefficients  $c_t^{[1]}, \ldots, c_t^{[n]}$ . The WF model outputs the local wind speeds  $v_w^{[1]}, \ldots, v_w^{[n]}$ , which are fed back to lookup