# Peer review of "Wind farm inertia forecasting accounting for wake losses, control strategies, and operational constraints"

_Wind Energy Science, 2025_

## Referee Comment (RC1)

**Summary**

The authors present a method to forecast the maximal guaranteed inertia provision by a wind farm. For that they combine:

- The results of an optimization problem to derive the maximum achievable inertia constant of a single WT for given combinations of curtailment level and wind speed under various restrictions
- 2. A forecast of the ambient wind conditions in the wind farm with a high spatial resolution
- 3. A wind farm of 12 WTs modelled as cP / cT LUTs, which are iteratively used in
- 4. A wake model of the wind farm.

The question of how much inertia a wind farm can support and how the support will affect the flow of the wind farm is highly relevant and new to the reviewer's knowledge. Insights into the constraints for inertia provision of single WTs in the different operating conditions are also highly relevant but have been discussed (less detailed) before. Different strategies for the curtailment of single WTs have been analysed before. The integration of such strategies and VSM into WT control have been shown before but is most likely still relevant and can still be optimized.

Overall it is not 100 % clear to me which parts of the paper are actually new and which parts have been published before. This is especially true for section 3.4 and reference Thommessen and Hackl 2024. If all of section 3.4 is new, it should be considered to break up the paper into two papers to allow a more detailed discussion: one on the optimization for individual WTs and a second for the specific effects in a WF. Just by looking at the number of pages and the level of discussion on WT effects compared to those of effects in a WF, the latter part might need some extra work. If it is decided to keep the paper as one, that should be reflected in the title.

However, despite the issues mentioned above, I can still recommend to publish the work as I consider it being relevant for the future discussion of the addressed issues.

**Style**

Please include a table with the most relevant parameters, e.g. by combining it with the list of abbreviations

While the general level of figures is good, figure 14 seems to be overloaded. Limiting the depicted WTs to exemplary ones or grouping them (e.g. first row, heavily influenced by wake effect, etc) would help to understand the differences between the similarly affected WTs in the WF.

Some important definitions of terms are missing, which the authors use heavily in the argumentation, e.g. initial ROCOF  $\rightarrow$  what time is meant? First second, first 10 ms?

Fast frequency response should be defined and differentiated from inertial response in section 2, it is assumed that the authors follow the definition by Eriksson et al.

Please consider choosing a different title. To me it suggests that the work is about control strategies and operational constraints of wind farms, these are (mainly) discussed on WT level in the paper. While this is of course also important for the wind farm, I had other expectations after reading the title (e.g. how to best distribute frequency support among the WTs in a WF). The abstract should read more like a summary of the paper at hand and less like an introduction. Furthermore, it is unclear why abbreviations are introduced there that are then reintroduced in section 1.1.

The conclusion should be reworked and be more focused on the findings in this paper, e.g the section on the background can be left out in my opinion.

**Citations**

Reference list must be reworked, harmonized and enhanced. For instance, NREL has an excellent documentation hence a reference like NREL: FORIS should not be used in this way

A lot of earlier important work is missing, e.g. but not limited to work on frequency support with WTs by G. Ramtharan, J.B. Ekanayake and N. Jenkins (2007) and NREL (J. Aho, L.Y. Pao, P. Fleming et al.) roughly 2012 to 2016.

It is unclear how the authors chose the citations (especially in section 1 and 2) given the tendency to self-citation while fundamental work of other authors is often missing here. One but not the only example: in line 20 the original source and not a self-citation must be used. It is good practice in science to give credit to the original authors of research.

There are numerous important statements without a (sufficient) reference given, e.g. on market design for inertia.

**Methodology and results**

Assumption of constant wind in the wind farm is arguable, especially due to the high non-linearity of the inertia constant to the wind speed

It is unclear whether the resolution in time of the wind speed forecast in the wind farm is sufficient for the intended forecast given the high non-linearity between wind speed and achievable inertia constant. Especially, how the minimum wind speed of the probability distribution is defined, whether it may further be reduced by turbulence and what effect that would have on the achievable inertia constant for single wind turbines. If turbulence is not included in the minima, the spatial distribution has a lesser smoothing effect on the sum of inertia constants than argued on page 7.

Has the influence of the MRS-based derating on WT loads been analysed yet?

Equation (15): what wind speed is used for that equation (rotor-effective?). Could it be a problem to measure / calculate that value in reality while operating at a reduced power setpoint?

Good to see that the authors use very challenging frequency events for their analysis. However,

- 1. Given that the authors use a control system, which increases the rated speed compared to the reference design, it should be shown, that the WTs do not have a risk of overspeed when a power reduction is needed.
- It would be interesting to see how the proposed controller performs when Hv is higher (e.g. e.g. at 6 or 7 m/s) → assumption would be that the effects on the aerodynamic performance are more relevant and thus more interesting for the discussion

Shouldn't the maximum rotor speed be added as a constraint for the optimisation problem when a power reduction is needed?

For the first test case, Hv is rather small. In fact, it is so small, that it may not be sufficient to allow a stable operation of the grid when all units had such an inertia constant. This is even more important when getting closer to full load operation. It would be great if the implications of the derived achievable inertia constants for the grid operation were (briefly) discussed.

It is unclear, why the inertial response is prioritized over fast frequency response even during the frequency recovery, i.e. IR is hampering the frequency recovery.

Might be a problem with the missing definition, however, it is unclear how the proposed control is able to reduce the initial ROCOF (in my understanding at the moment of the frequency disturbance) when it is stated in line 330 that measurements of the voltage (and most likely the calculation of the frequency is needed). Please clarify it.

Figure 4 shows that the optimal rotor speed is maximum 20 % above the minimal rotor speed during the transition between region I and II. Why is the minimum rotor speed not the constraint for the lowest wind speeds (up to appr. 4 m/s) in figure 11 b, which show a reduction of 20 % compared to MPPT for these wind conditions?

Figure 18: in what cases is the sum of  $H_V$  in a WF underestimated when wake effects are neglected?

It is discussed whether or not a power reduction should be allowed during the speed recovery phase. While this is certainly an important issue, it is very important to compare similar cases. At least for some time, the Quebecois grid code (which usually the relevant case for Godin et al.) allowed very long recovery periods (in the range of 30 s) for moderate power increases. Thus only very small power reductions are needed during the recovery. This is very different from the simulated case study with higher power increases and with a much shorter recovery period.

---

## Author Comment (AC1)

**Reply to the reviewers' comments for the paper "Wind farm inertia forecasting accounting for wake losses, control strategies, and operational constraints"**

The authors would like to thank the two reviewers for their time and for the useful feedback. All the inputs that they provided have contributed to the improvement of the paper.

A list of point-by-point replies to the reviewers' comments is reported in the following. The reviewer's comments are in black, and our replies in blue.

We have taken the opportunity of this deep revision to improve readability and understanding by adding explanations and bibliographical references.

A revised version of the manuscript is attached to the present reply, with additions highlighted in blue and corrections/deletions marked in red.

The authors

**Reviewer 1**

**Summary**

The authors present a method to forecast the maximal guaranteed inertia provision by a wind farm. For that they combine:

> 1. The results of an optimization problem to derive the maximum achievable inertia constant of a single WT for given combinations of curtailment level and wind speed under various restrictions
> 2. A forecast of the ambient wind conditions in the wind farm with a high spatial resolution
> 3. A wind farm of 12 WTs modelled as cP / cT LUTs, which are iteratively used in
> 4. A wake model of the wind farm.

The question of how much inertia a wind farm can support and how the support will affect the flow of the wind farm is highly relevant and new to the reviewer's knowledge. Insights into the constraints for inertia provision of single WTs in the different operating conditions are also highly relevant but have been discussed (less detailed) before. Different strategies for the curtailment of single WTs have been analysed before. The integration of such strategies and VSM into WT control have been shown before but is most likely still relevant and can still be optimized.

Overall it is not 100 % clear to me which parts of the paper are actually new and which parts have been published before.

As correctly pointed out by the reviewer, the present submission -as most papers- builds on some of our previous works, and specifically on Anand et al., 2024; Thommessen and Hackl, 2024. However, we are a bit surprised by this comment. In fact, the whole Sect. 1.3 was very specifically dedicated to highlighting the novelty of this article. We would like to report here the

beginning of Sect. 1.3, because we think that it leaves few doubts about what was missing in the literature and what is new in our paper:

*"To the best of our knowledge, a generic approach for evaluating the maximum deliverable inertia from WFs for grid-forming control is still missing. Moreover, the methodology for predicting WF inertia based on operation plans has not yet been discussed, although this is key for the reliable and efficient operation of future power systems. Furthermore, even though the intra-farm turbine-to-turbine interactions have a huge influence on the local inflow at the turbines, they have largely been ignored in the existing studies evaluating inertia provision capability. Thus, this paper proposes a novel and generic approach for WF inertia forecasting. This holistic methodology considers weather prediction models, WF flow effects due to wake interactions among the WTs, control strategies, and operational constraints to predict the maximum deliverable inertia at the WT and WF levels."*

In that same section, for maximum clarity, we even highlighted the contributions of the paper using a bulleted list.

This is especially true for section 3.4 and reference Thommessen and Hackl 2024. If all of section 3.4 is new, it should be considered to break up the paper into two papers to allow a more detailed discussion: one on the optimization for individual WTs and a second for the specific effects in a WF. Just by looking at the number of pages and the level of discussion on WT effects compared to those of effects in a WF, the latter part might need some extra work.

Thank you for the suggestion to split the paper, but we strongly recommend against this solution, because of the following reasons.

The paper essentially builds on three elements: 1) forecasting at the site level, 2) bringing the forecasts at the level of the single turbine, and 3) finally computing the maximum inertia that a turbine can generate based on its predicted inflow conditions.

First, splitting these three elements into two papers would break the strong connections that actually exist among them. Indeed, it is one of the key contributions of this work to bring -for the first time- these three elements together.

Second, farm effects appear to occupy less space in the paper simply because they are accounted for using well-known engineering wake models. While these models are widely used, we provided the necessary bibliographical references, where the readers can find all necessary background information. A separate paper only devoted to farm effects would be either very short, or present information that is widely available.

Third, another key contribution of this work is the formulation of the maximum inertia provision of a single wind turbine, accounting for its operational limits. This part is absolutely novel, and never reported before. In order to be comprehensible by a reader, this new formulation requires significant background information. For this purpose, in Sect. 3.4 we have used a more generic and simplified representation than the one developed in Thommessen and Hackl 2024. We strongly believe that, without this level of detail, it would be extremely difficult for a reader to have a clear understanding of this material.

We believe that a split of the paper would not improve the presentation. Therefore, we respectfully ask to allow us to maintain the current structure, which was based on a careful analysis of the material, with a focus on the interests of the readers.

If it is decided to keep the paper as one, that should be reflected in the title.

We have changed the title to "Wind farm inertia forecasting accounting for wake losses, turbine-level control strategies, and operational constraints". In our opinion, the title clearly indicates the content of the paper, while taking into account that the title has to be relatively short.

However, despite the issues mentioned above, I can still recommend to publish the work as I consider it being relevant for the future discussion of the addressed issues.

Thank you for the comment.

**Style**

Please include a table with the most relevant parameters, e.g. by combining it with the list of abbreviations.

Thank you for the comment.

If you refer to a list of acronyms and symbols, this can be found in Appendix J.

If you refer to modeling parameters, then most of their values are stated in the text and/or are visible in the figures (e.g., Fig. 3, 4, 7, 8). Additionally, the WT model is based on a reference design, whose complete data is available in the public domain in bibliographical references that have been cited in our paper.

For completeness of information, we have now added Figure 7, which shows the WF layout.

We would like to stress that, although demonstrated with reference to a specific WF and WT type, the proposed approach is generic and can be applied to arbitrary WFs and/or WTs.

While the general level of figures is good, figure 14 seems to be overloaded. Limiting the depicted WTs to exemplary ones or grouping them (e.g. first row, heavily influenced by wake effect, etc) would help to understand the differences between the similarly affected WTs in the WF.

Thank you for this comment. In our opinion, it is nice to see (or at least have an impression of) all the raw data in Fig. 14, before considering mean values only in the subsequent figures. Further, the mean values (red lines) in Fig. 14 capture the general trend of all individual curves. Of course, understanding the wake effects in detail would be nice, but this is beyond the scope of this paper. However, we have added Fig. 7, which illustrates the WF layout as well as the resulting wake interactions within the WF under exemplary wind conditions.

Some important definitions of terms are missing, which the authors use heavily in the argumentation, e.g. initial ROCOF → what time is meant? First second, first 10 ms?
Fast frequency response should be defined and differentiated from inertial response in section 2, it is assumed that the authors follow the definition by Eriksson et al.
Thank you for the comments. A corresponding footnote has been added in Sect. 1.1 (line 24) that defines the initial ROCOF and differentiates inertia from (fast) frequency response (see also Fig. 2.4 below from AEMC, 2017).

**Figure 2.4    Timeline for inertia and fast frequency response**

[Figure]

Please consider choosing a different title. To me it suggests that the work is about control strategies and operational constraints of wind farms, these are (mainly) discussed on WT level in the paper. While this is of course also important for the wind farm, I had other expectations after reading the title (e.g. how to best distribute frequency support among the WTs in a WF).

We have changed the title to "Wind farm inertia forecasting accounting for wake losses, turbine-level control strategies, and operational constraints". We believe that the present version clearly indicates that the control aspects are addressed at the turbine level, while foresting is performed accounting for wakes effects, therefore resulting in farm-level predictions.

The abstract should read more like a summary of the paper at hand and less like an introduction. Furthermore, it is unclear why abbreviations are introduced there that are then reintroduced in section 1.1.

Thank you for the comment. The considered topic is rather new, or at least not well known. Thus, we added a short introduction and motivation in the abstract before summarizing the proposed approach. We have expanded the abstract to accommodate your request and to highlight more the contributions of this work. According to the journal style, acronyms have to be (re-)introduced after the abstract.

The conclusion should be reworked and be more focused on the findings in this paper, e.g the section on the background can be left out in my opinion.

The conclusion in Sect. 5 summarizes the motivation, the methodology, and the findings of this work. Additionally, Section 5 provides an outlook on future work. Whereas Section 4 presents the simulation results and discusses the findings based on numbers (quantitative analysis), Section 5 provides a rather qualitative summary of the findings. In our opinion, the conclusion contains all key findings of the paper and is not too long, although it contains some background info. If there are any key findings missing in the conclusion, please let us know.

**Citations**

Reference list must be reworked, harmonized and enhanced. For instance, NREL has an excellent documentation hence a reference like NREL: FORIS should not be used in this way.

Thank you for the comment. The reference has been expanded.

A lot of earlier important work is missing, e.g. but not limited to work on frequency support with WTs by G. Ramtharan, J.B. Ekanayake and N. Jenkins (2007) and NREL (J. Aho, L.Y. Pao, P. Fleming et al.) roughly 2012 to 2016.

Aho et al. (2014) only consider secondary frequency regulation. G. Ramtharan et al. (2007) only consider grid-following control with (fast) frequency response, which cannot limit the initial ROCOF, whereas our work considers grid-forming control with VSM inertia provision. Ramtharan et al. (2007) and Aho et al. (2014) propose WT derating by shifting the operating point towards the right of the maximum power curve to increase the kinetic energy reserve, similar to the maximum rotation strategy (MRS) considered in this paper. These references have now been added to the "Background and fundamentals" in Sect. 2 (lines 140f.): "For inertia provision, derating based on the maximum rotation strategy (MRS) additionally increases the WT kinetic energy reserve (Ramtharan et al., 2007; Aho et al., 2014; Meseguer Urban et al., 2019; Thommessen and Hackl, 2024)."

It is unclear how the authors chose the citations (especially in section 1 and 2) given the tendency to self-citation while fundamental work of other authors is often missing here. One but not the only example: in line 20 the original source and not a self-citation must be used. It is good practice in science to give credit to the original authors of research.

Thank you for the comment. A new citation (ENTSO-E, 2016) has been added to the considered sentence in line 23f.: "Additionally, the initial ROCOF also increases due to increasing power system imbalance (ENTSO-E, 2016; Thommessen and Hackl, 2024)." Note that the original source (ENTSO-E, 2016) or other sources often use (over-)simplified models of the power system dynamics. Thus, the work (Thommessen and Hackl, 2024), which uses advanced modeling (electromagnetic transient modeling), was also added to make clear that the statement is not only true for (over-)simplified models but also for more sophisticated models or reality.

There are numerous important statements without a (sufficient) reference given, e.g. on market design for inertia.

Thank you for the comment. The inertia market designs and implementations are still ongoing. To the best of the authors' knowledge, there is no further official information available about the considered future German inertia market beyond the cited documents. Further, "Ghimire et al. (2024) present a review of existing functional specifications and testing requirements of grid-forming offshore WFs" (see lines 47f. in Sect. 1.1).

**Methodology and results**

Assumption of constant wind in the wind farm is arguable, especially due to the high non-linearity of the inertia constant to the wind speed.
It is unclear whether the resolution in time of the wind speed forecast in the wind farm is sufficient for the intended forecast given the high non-linearity between wind speed and achievable inertia constant. Especially, how the minimum wind speed of the probability distribution is defined, whether it may further be reduced by turbulence and what effect that would have on the achievable inertia constant for single wind turbines. If turbulence is not included in the minima, the spatial distribution has a lesser smoothing effect on the sum of inertia constants than argued on page 7.

Thank you for the comments. The wind speed is not assumed to be constant in general (see Fig. 14). Clearly, our work considers wake effects, so the wind speed is far from spatially constant, and it is actually different for each individual WT in the WF (see Fig. 15); the inclusion of wake

effects is indeed one of the key contributions of this paper. We assume a constant wind speed only during the short time interval of the inertial response to a severe ROCOF, which is reasonable in our opinion, as discussed in the beginning of Sect. 3 (lines 164-172). Note that Aho et al. (2014), whose work you mentioned previously, similarly expect that "the spatial filtering of aggregating the power of multiple turbines over a larger geographical area would reduce the effective variability in the wind resource."

Regarding time resolution, we agree that future work could verify the effects of a higher sampling frequency. However this, as many other corollary aspects, are parametric studies that would add little to the essence of the present work.

Has the influence of the MRS-based derating on WT loads been analysed yet?

Thank you for the comment. Aho et al. (2014) observe a reduction of damage equivalent loads during derating, which aligns with the results in Sect. 4.1 in this work. Thus, a corresponding sentence with citation has been added at the end of Sect. 4.1 (line 455).

Equation (15): what wind speed is used for that equation (rotor-effective?). Could it be a problem to measure / calculate that value in reality while operating at a reduced power setpoint?

Thank you for the comment. Equation (15) uses the rotor-effective wind speed, as is done throughout this work. There is ample literature on the estimation of such quantity, and we have now added a reference to a highlycited review paper (Slotani et al., 2013) on this topic just before Equation (15. However, Equation (16) limits the derating power setpoint to the MPPT setpoint to prevent excessive rotor deceleration caused by wind speed estimation errors. Thus, rotor speed cannot decrease below its MPPT value.

Good to see that the authors use very challenging frequency events for their analysis. However,

1. Given that the authors use a control system, which increases the rated speed compared to the reference design, it should be shown, that the WTs do not have a risk of overspeed when a power reduction is needed.
   Thank you for the comment. The rated rotor speed is a few percent higher than in Bortolotti et al. (2019), since our work considers MPPT below rated wind speed, whereas Bortolotti et al. (2019) include a tip speed constraint below rated wind speed. However, the resulting rated speed in our work is still more than 5 % lower than the maximum rotor speed limit in Bortolotti et al. (2019), see the second paragraph in Sect. 3.4 (lines 269-274). If this (over-)speed reserve is not sufficient for sudden generator torque reductions, a more aggressive tuning of the pitch control or a higher generator speed rating is needed. However, those investigations are, in our opinion, out of scope of this work.

2. It would be interesting to see how the proposed controller performs when $H_v$ is higher (e.g. e.g. at 6 or 7 m/s) → assumption would be that the effects on the aerodynamic performance are more relevant and thus more interesting for the discussion.
   Thank you for the comment. We already included two examples with different $H_v$-values in Figs. 9-10. In our opinion, including more examples would be too much, since the paper is already rather long.

Shouldn't the maximum rotor speed be added as a constraint for the optimisation problem when a power reduction is needed?

Thank you for the comment. While both power increase and decrease are relevant, this work considers a (generator) power increase to be the worst-case scenario due to the risk of critical rotor deceleration. It is assumed that power decreases can be managed by pitch control, which limits the rotor acceleration. This is reasonable, as pitch control is also responsible for speed limitation during other events with significant power reduction, such as low-voltage ride-through.

For the first test case, $H_v$ is rather small. In fact, it is so small, that it may not be sufficient to allow a stable operation of the grid when all units had such an inertia constant. This is even more important when getting closer to full load operation. It would be great if the implications of the derived achievable inertia constants for the grid operation were (briefly) discussed.

Thank you for the comment. In our opinion, the implications are clear from the introduction in Sect. 1. If wind resources cannot provide enough inertia in MPPT mode, derating can increase the inertia provision, or other resources need to provide more inertia for stable grid operation. Clearly, considering Eq. (1), ensuring lower worst-case power imbalances or allowing for higher ROCOFs would reduce the general demand for power system inertia.

It is unclear, why the inertial response is prioritized over fast frequency response even during the frequency recovery, i.e. IR is hampering the frequency recovery.

Thank you for the comment. An explanation of this aspect has now been added in Section 3.4.5 and in the new Appendix I: "[...] For our control strategy, this implies that the inertia provision based on additional kinetic energy extraction is prioritized over droop control; see Appendix I for details." (see line 391-399)

Might be a problem with the missing definition, however, it is unclear how the proposed control is able to reduce the initial ROCOF (in my understanding at the moment of the frequency disturbance) when it is stated in line 330 that measurements of the voltage (and most likely the calculation of the frequency is needed). Please clarify it.

Thank you for the comment. For clarity, the sentence at this place has been expanded to: "Grid voltage measurements are required to determine $\Omega_s$ for VSM damping torque calculation." (line 336). That means the calculation of the frequency is only needed for the VSM damping. In contrast, the electromagnetic feedback for VSM control does not require the calculation of the frequency. For clarity, the corresponding description labels in Fig. 5 have been expanded to "Electromagnetic feedback (physics)", "VSM (control)", and "Power system dynamics (ROCOF event)". The VSM feedback is directly calculated based on the estimated torque or power, obtained by current and grid voltage measurements, with no need to calculate the grid frequency, the grid angle $\phi_s$ or load angle $\delta$ (Thommessen and Hackl, 2024). This is also stated in Sect. 3.4.3 (line 333f.): "Due to unknown $\phi_s$ or $k_e$, the VSM controller calculates the torque or power feedback directly based on current and grid voltage measurements." When the frequency or system angle $\phi_s$ changes, the load angle or output power changes instantaneously, i.e., without the (fast) frequency response delay of grid-following controllers. Thus, the inertial response of the proposed grid-forming VSM control reduces the initial ROCOF, similar to real grid-connected SMs.

Figure 4 shows that the optimal rotor speed is maximum 20 % above the minimal rotor speed during the transition between region I and II. Why is the minimum rotor speed not the constraint for the lowest wind speeds (up to appr. 4 m/s) in figure 11 b, which show a reduction of 20 % compared to MPPT for these wind conditions?

Thank you for the comment. We are not sure if we fully understand your first sentence, where you refer to Fig. 4. We agree that the rotor speed is higher than its optimal value during the transition interval between regions I and II, which is evident in Fig. 8e at the lowest wind speed, where the tip speed ratio is higher than its optimal value of 8.5. Figure 12b shows the rotor speed nadir or deviation to the MPP speed. The MPP speed is defined as the initial (steady-state) speed in MPPT mode. For clarity, the corresponding definition "$\Omega_{mpp} := \Omega_0$ for $P_{set} = 1$" has been added to the description of Fig. 12b in the text (line 567). Thus, the data points for $P_{set} = 1$ in panels (a) and (b) of Fig. 12 are equal. The minimum rotor speed constraint is active at the red data points with low wind speed and low derating. For the green data points, the torque rate constraint is active, i.e., due to the low speed, a high torque increase is needed to achieve the desired output power for inertia provision. Although the speed deviation is up to ca. 18 % for the green data points, the speed nadir stays above the minimum speed due to sufficient initial speed. Thus, the minimum rotor speed is not relevant for most operating points. In other words, limiting the torque rate implies rotor speed protection for most operating points.

Figure 18: in what cases is the sum of $H_v$ in a WF underestimated when wake effects are neglected?

Section 4.3 discusses uncertainties in both the ambient wind forecast and the wake modeling. Due to the nonlinearity of the mapping from wind speed to inertia provision (see Fig. 11), it depends on the operating point whether a positive or negative wind speed forecast error leads to inertia over- or underestimation. However, the discussions in Sect. 4.3 give more insights, e.g. (lines 637ff.): "In Figures 19a and 19c, the WF inertia forecast error distributions of the "Simple Opt" and "No Wake" variants are shifted to the left compared with the proposed variant due to the inertia underestimation in Fig. 18a. At the samples with $\Delta H_{v,max} > 0$ in 19a, the wind forecast errors overcompensate for the modeling errors $\Delta H_{v,max} < 0$ in Fig 18a."

It is discussed whether or not a power reduction should be allowed during the speed recovery phase. While this is certainly an important issue, it is very important to compare similar cases. At least for some time, the Quebecois grid code (which usually the relevant case for Godin et al.) allowed very long recovery periods (in the range of 30 s) for moderate power increases. Thus only very small power reductions are needed during the recovery. This is very different from the simulated case study with higher power increases and with a much shorter recovery period.

Thank you for the comment. The considered worst-case scenario is defined in the German grid code (VDE, 2024b). This scenario may not represent a realistic frequency event, but it can be considered as a worst-case test to verify the inertia provision. Godin et al. (2019) consider a frequency event with less severe ROCOFs, which requires a lower power increase $\Delta P$ over a longer period $\Delta t$, before reaching the frequency nadir. However, the kinetic energy extraction $\Delta E = \Delta P \Delta t$ may be similar to the one in our work due to lower $\Delta P$ but higher $\Delta t$ in their work. Although Godin et al. (2019) consider a slower frequency recovery afterwards, the rotor speed recovery is similar in principle. Note that the rotor speed recovery can take much more time than the frequency recovery (see Fig. 9). Of course, it would be nice to compare our case with a more similar case based on measurements, but we could not find a corresponding publication, nor do we have access to such measurements.

For clarity, the text in Appendix G has been modified and expanded.

**Reviewer 2**

The authors present a method to predict the grid-forming capabilities of wind turbines on a wind farm, taking wind turbine control strategies and operating limits into account. Previous work by some of the authors (Anand et al., 2024) forms a basis for forecasting inflow conditions at each wind turbine in a wind farm, incorporating wake effects in the wind farm. The resulting flow conditions are then used in predicting the grid-forming capabilities of the wind turbines. The (Thommessen and Hackl, 2024) paper is cited numerous times throughout the paper in the presentation of the various parts needed for predicting the inertia that wind turbines can provide, and I am not completely sure of the new contributions the authors are providing in this current paper. It would be helpful for the authors to more clearly state what has been done in the past, and what their *new* contributions are with this paper. Are they assembling many pieces from past works, which has never been done before? Are some pieces novel and were required to be developed as the authors were putting together the overall inertia forecasting capability?

Please see our first response to the first reviewer above.

While I enjoyed the paper overall, the paper is quite complex and at times difficult to understand and follow. There are often references within the same sentence to an appendix at the end, a figure several pages later in the paper, and a figure that was previously shown several pages earlier in the paper. This cross-referencing to multiple other places in the paper sometimes made it difficult to follow.

Thank you for the comment. However, we could not find a reference in the text to a "figure several pages later in the paper". Could you please indicate which reference you mean? We could only find the following two exceptions:

- in footnote 4 on page 16, which refers to the reference frequency event shown Fig. 10 to explain the details of inertia provision quantification, and
- in the enumeration list at the beginning of Section 4.3, which refers to all figures shown in this subsection to give an overview of the results, which are presented subsequently.

We think that these two exceptions are acceptable and helpful for the reader. All other figure references refer to a figure that appears immediately before or after the reference (on the previous, same, or next page), or that has been introduced previously. Cross-references to the appendix have been added instead of including all details in the main part of the paper to maintain the reading flow. All necessary details are still available in the appendix. Cross-references to previous figures underpin the argumentation and are needed to understand the interconnections between the different methods and results.

In Section 4, the authors present a case study for a wind farm with 12 wind turbines. What is the layout of the wind turbines? This would be useful, for instance, in interpreting the individual wind turbine curves in Figure 14(a).

Thank you for the comment. A new figure has been added (Fig. 7) that shows the layout. The corresponding text describes the figure and explains the wake effects for an exemplary relevant ambient wind condition.

A few more detailed comments and suggestions for improvement:

1. Below Equation (9), the authors state that "The i-th constraint is considered active if $c_i = 0$, or inactive if $c_i > 1$." Should the "1" be a "0"?

   Yes, thank you for spotting this mistake, which has now been corrected.

2. When Figure 2 first appears, the authors state at the beginning of Section 3.4 that "Figure 2 depicts the overall WT modeling and control used in this work." Figure 2 actually depicts a lot more than the overall WT modeling and control used. It would be useful to the reader if the authors provided an overall description of the rather complex and encompassing Figure 2 when the figure is first introduced, before the various sections that then focus on discussing particular parts of the figure in further detail.
   This is an test. Why does grammar not work?

   Thank you for the suggestion. At the beginning of Sect. 3.4 (lines 266-268), we have added a short introduction to the key control concept and the corresponding blocks/signals in Fig. 2. The subsequent sections explain Fig. 2 step by step in more detail.

3. It would be useful to point out to the reader early on that Appendix J provides a listing of the nomenclature used throughout the paper. I don't believe Appendix J is ever mentioned in the main text, and at least for me, I didn't see that Appendix until well into the paper. Indeed, there are many symbols used and it quickly became confusing; and it would have been useful to know of Appendix J before I discovered it much later on my own.

   Sorry for the inconvenience. We have added a corresponding footnote above equation (1) in line 111. Note that this footnote refers to Appendix K, which corresponds to Appendix J in the old paper version, since Appendix I has been added in the new paper version.

4. On page 14, lines 319-320, the authors state "the VSM mechanical model is based on a one-mass model with virtual inertia constant $H_v$ in Eq. (13)." When I look at Eq. (13), I don't see any $H_v$, and as a result I am confused and am not sure I am understanding the statement correctly.

   For clarity, we have expanded the text to "the VSM mechanical model is based on a one-mass model with virtual inertia $H_v$ instead of physical inertia $H_R$ in Eq. (13)." (line 325f.)

5. On page 21, lines 498, in the discussion of Fig. 8, the symbol "beta_min" is used. From Fig. 8, panel (c), it appears that this beta_min is 1.1 degrees (green curve).  Since there are several other curves in Fig. 8(c) with beta values *smaller* than 1.1 degrees, I would suggest using "beta_finepitch" or some other symbol other than "beta_min" since beta_min is clearly not the minimum beta.

   Thank you for the suggestion. For clarity, we have replaced "beta_min" with "beta^star" in this place (line 508):

   > "[...] the power coefficient starts at its maximum value due to MPPT, i.e. (beta, lambda) = (beta^star, lambda^star) [...]".

   Correspondingly, we have also replaced "beta_min" with "beta^star" in the paragraph after next (line 523):

   > "[...] , and (ii) active pitch control, i.e., beta($t_0$) > beta^star in Fig. 9."

   Note that in this work, the lower limit "beta_min" is considered as a function of the tip speed ratio "lambda". This is mentioned several times, e.g., just before Sect. 3.4.1 (line 285), in the 2nd paragraph of Sect. 4.1 (line 443), and in equation (B3). A corresponding lookup table (LUT) for the lower limit is implemented in the pitch reference controller, as shown in the plot below. This beta_min-LUT is obtained by maximizing the power coefficient c_p = c_p(lambda, beta) for given tip speed ratios lambda (see also line 443). However, we think that this LUT or further implementation details are beyond the scope of this work.

[Figure]

The optimal pitch angle (at the MPP) is "beta^star = 1.1 degrees", which equals the lower limit "beta_min" at the optimal tip speed ratio "lambda^star = 8.5" (as shown in the plot above). The optimal values are also stated in the caption of Fig. 3 in the paper. In Fig. 9, panels (c) and (f) show beta-values *smaller* than 1.1 degrees due to (slightly) decreased tip speed ratio, i.e., lambda < lambda^star.

6. On page 21, lines 500-501, the authors state that "After the ROCOF changes from negative to positive at $t_{s,min}$ = 3.5s, ..." Should that be 2.5s instead? Indeed, later in the same paragraph (line 504), the authors indicate that "$t_{s,min}$ = 2.5s".
Yes, thank you for spotting this mistake, which has now been corrected (see line 511).

7. In Figure 10, on the right plot, the legend has the ordering of Omega_min, dot{M}_{e,max}, M_{e,max}, P_{e, max}. The ordering is then changed in the legends in the left plots of Figures 11 and 12. I would suggest using a common ordering for the legends in all of these plots.
We do not see any issues with the legends, as the color coding is consistent across all plots (see Fig. 11-13). The legend order reflects the sequence in which the markers are plotted (newly plotted markers appear in the foreground). The legend may vary depending on the optimal 3D-view and the plotting order.

8. On page 28, line 607, the authors indicate "$P_{set}$ = 98% (panel b)" but panel (b) has a plot title indicating "$P_{set}$ = 0.96". It's not clear which is the actual $P_{set}$.
Thank you for spotting this mistake. The value in the text was wrong and has now been corrected to 96% (see line 617).

9. Sometimes closing-quotes show up instead of opening-quotes when quotes are used around "Det", "Comb Min", "Simple Opt", "No Wake", "Deterministic" in Figures 16-18.
Thank you for spotting these mistakes, which have now been corrected (see Fig. 17-19).

10. If possible, please move the legends so that they do not cover the results "bars" in Figures 17-18. It seems there is enough white space in the plots to move the legends to more "optimal" locations that don't cover any of the results actually being shown.
Thank you for the suggestion, but –unless we reduced the legend size, which would have hindered readability– we could not find better locations. However, please notice that the legends do not cover or hide any results, since the top ends of the bars are visible in all plots.

**Wind farm inertia forecasting accounting for wake losses, turbine-level control strategies, and operational constraints**

Andre Thommessen[1], Abhinav Anand[2], Carlo L. Bottasso[2], and Christoph M. Hackl[1]

[1]Laboratory for Mechatronic and Renewable Energy Systems (LMRES), Hochschule München (HM) University of Applied Sciences, 80335 Munich, Germany
[2]Wind Energy Institute, Technical University of Munich (TUM), 85748 Garching bei München, Munich, Germany

**Correspondence:** Andre Thommessen (andre.thommessen@hm.edu)

**Abstract.** Future inverter-based resources (IBRs) must provide grid-forming functionalities to compensate for the declining share of conventional synchronous machines (SMs) in the power generation mix. Specifically, decreasing power system inertia poses a significant challenge to grid frequency stability, as system inertia limits the rate of change of frequency (ROCOF). Conventional grid-following control decouples the physical inertia of wind turbines (WTs) from the grid frequency. Novel grid-forming control methods, such as virtual synchronous machine (VSM) control, provide (virtual) inertia to the system, e. g. by extracting kinetic energy from WTs. Since the grid-forming capability of IBRs depends on volatile operating conditions, future market designs will remunerate inertia provision based on its availability. Thus, estimating grid-forming capabilities of WTs and forecasting inertia of wind farms (WFs) are of interest for both WF and system operators.

In this paper, we propose a method for forecasting WF inertia that accounts for wake effects and WT characteristics. A wake model estimates individual inflow conditions for each WT in the WF based on forecasted site conditions. These inflow conditions are crucial for predicting the grid-forming capabilities of each WT. Under varying inflow conditions and derating power setpoints, we simulate the WT inertial responses to a reference frequency event. Taking WT control strategies and operating limits into account, an optimization algorithm computes the maximum feasible inertia provision at the WT and WF level. The proposed approach is demonstrated in a simulation environment, and the results also include a quantification of the uncertainties due to both wind forecasting and wake modeling errors.

**1 Introduction**

**1.1 Motivation and problem statement**

Imbalances between power generation and demand result in frequency events. Thus, generation or protection units must rapidly compensate for power imbalances to keep the grid frequency within admissible limits (ENTSO-E, 2021). Following an imbalance event, the power system inertia limits the rate of change of frequency (ROCOF) (ESIG, 2022). Historically, the rotating masses of directly coupled synchronous machines (SMs) provided sufficient inertia to limit the ROCOF. However, with the decreasing share of SMs and the increasing share of inverter-based resources (IBRs) in the overall generation mix, the power system inertia is decreasing (ENTSO-E, 2021). Additionally, the initial ROCOF also increases due to increasing power system

imbalance (ENTSO-E, 2016; Thommessen and Hackl, 2024).[1] Worst-case frequency events are caused by faults that split the system into subsystems due to a sudden loss of electrical import or export power (ENTSO-E, 2021). Furthermore, increasing transmission capacities, such as high voltage direct current (HVDC) links, may lead to even higher future worst-case power imbalances during system splits (ENTSO-E, 2021). Consequently, IBRs must provide inertia to limit the ROCOF and to avoid blackouts in future power systems (ENTSO-E, 2021).

Wind farms (WFs) can support grid frequency by supplying inertia and fast frequency response through the rotating masses of the wind turbines (WTs) and by providing reserves (if available). However, conventional grid-following control decouples the "physical" inertia of WTs from the grid frequency and thus can *not* provide inertia to the grid (Bossanyi et al., 2020). Advanced grid-following control such as "WindINERTIA" control from General Electric (Clark et al., 2010), or the "inertia emulation (IE)" control from ENERCON (Godin et al., 2019), can temporarily extract kinetic energy reserves to support grid frequency. However, this so-called "synthetic" inertia cannot limit the instantaneous or initial ROCOF subject to a system disturbance (AEMC, 2017; ENTSO-E, 2021; ESIG, 2022). On the contrary, new grid-forming control methods for IBRs, such as virtual synchronous machine (VSM) control, provide (virtual synchronous) inertia that limits the initial ROCOF (ESIG, 2022; Bossanyi et al., 2020; Rodriguez-Amenedo et al., 2021; Thommessen and Hackl, 2024; Ghimire et al., 2024). Consequently, future WFs should integrate grid-forming control to provide inertia and fast frequency response. However, this is not only a WT control problem, because what a WT can deliver ultimately depends on the intra-farm wake-dominated flows that develop within WFs.

New grid codes and market incentives for grid-forming technologies are paving the way for the stability of future power systems (ESIG, 2022). Accordingly, system operators are transitioning towards the procurement of inertia provision by grid-forming technologies. For instance, due to the high penetration of IBRs in Great Britain, the National Grid Electricity System Operator already defines technical requirements for grid-forming technologies in the grid code and includes grid-forming capability as a market product (ESIG, 2022). Similarly, German system operators plan to establish an inertia market and to remunerate inertia provision based on its availability (Bundesnetzagentur, 2024). Accordingly, the new German specifications (VDE, 2024a) already define technical requirements for grid-forming control and inertia provision. Ghimire et al. (2024) present a review of existing functional specifications and testing requirements of grid-forming offshore WFs. Hu et al. (2023) design an inertia market to ensure sufficient system inertia and analyze its impact on the power generation mix. Their results show that investing in wind resources with virtual inertia facilities is more cost-competitive than substituting wind resources with thermal generators, not to mention the improved environmental impacts.

System inertia monitoring and forecasting are essential to ensure adequate inertia provision. More precisely, system operators need to quantify the minimum required system inertia to survive worst-case system splits and need to procure sufficient inertia provision. Given the uncertainty and variability associated with renewable energy sources, system operators need inertia forecasting to ensure that sufficient inertia is available at any time. Similarly, WF operators need WF inertia forecasting to participate in future availability-based inertia markets. In particular, WF inertia forecasting enables reliable and profitable
* * *
[1]The initial ROCOF is the mean ROCOF over a time window of a few hundred milliseconds after an event, during which only inertia can limit the drop in frequency, before other technologies can activate a (fast) frequency response (AEMC, 2017, p. 13, Fig. 2.4; Thommessen and Hackl, 2024, p. 285).

[revised manuscript text omitted]

- analyzing WT dynamics and relevant operating limits by simulating the inertial response to a reference frequency event,

- integrating VSM control and modifying WT control for inertia provision and fast frequency response,

- demonstrating the proposed approach for evaluating and forecasting deliverable inertia at the WT and WF levels,

- comparing the proposed approach with simplified ones for estimating WT grid-forming capabilities, and

- evaluating the impact of forecast uncertainty, wake effects, control strategies, and WT model inaccuracies on WF inertia forecasting.

The rest of this paper is organized as follows. Section 2 presents the necessary background and fundamentals regarding system inertia, ROCOF, and inertia provision by WTs using the VSM concept. Section 3 presents the proposed approach in detail. This includes all the necessary steps for WF inertia forecasting: (i) WF ambient wind conditions forecast in Sect. 3.1, (ii) local WT operating points prediction in Sect. 3.2, and (iii) mapping of all operating points, given by local wind inflow conditions and operational setpoints, to the WF grid-forming capability in Sects. $3.3 - 3.4$. Section 4 presents a case study for a WF with twelve WTs and discusses the results, including the WT steady states, the WT inertial response to a reference frequency event, and the WF hour-ahead inertia forecasting. Finally, Sect. 5 summarizes the entire work and offers concluding remarks, including outlook for future work.

**2   Background and fundamentals**

The initial ROCOF immediately after a system power imbalance $\Delta P_\mathrm{s}$ between mechanical system power $P_\mathrm{m,s}$ and electrical system power $P_\mathrm{e,s}$ can be approximated by a one-mass model (ENTSO-E, 2020; Thommessen and Hackl, 2024), written as[2]

$$\dot{\Omega}_\mathrm{s} = \frac{\Delta P_\mathrm{s}}{2H_\mathrm{s}} := \frac{P_\mathrm{m,s} - P_\mathrm{e,s}}{2H_\mathrm{s}}, \tag{1a}$$

$$\text{where} \quad H_\mathrm{s} := \frac{E_\mathrm{kin,s}}{p_\mathrm{s,R}} := \frac{\frac{1}{2}\Theta_\mathrm{s}\omega_\mathrm{s}^{2}}{\sum p_\mathrm{R}}, \quad \Theta_\mathrm{s} := \sum \Theta \quad P_\mathrm{m,s} := \frac{\sum p_\mathrm{m}}{p_\mathrm{s,R}}, \quad \text{and} \quad P_\mathrm{e,s} := \frac{\sum p_\mathrm{e}}{p_\mathrm{s,R}}. \tag{1b}$$

The system inertia constant $H_\mathrm{s}$ (in s) is the system kinetic energy $E_\mathrm{kin,s}$ (in W s) normalized to the rated system power $p_\mathrm{s,R}$ (in W), defined as the sum of rated power $p_\mathrm{R}$ of all (V)SMs. The system moment of inertia $\Theta_\mathrm{s}$ (in $\mathrm{kg\,m^2}$) is defined as the sum of the moment of inertia $\Theta$ of all (V)SMs. This includes (V)SMs at the generation side but also at the demand or load side, i. e.
* * *
[2]For the nomenclature used in this work, see Appendix K.

(V)SMs provide inertia in both generator and motor mode. The system angular velocity is $\omega_s = 2\pi f_s \approx 2\pi f_{s,R}$ with system frequency $f_s$ (in Hz) of all synchronously rotating masses and rated system frequency $f_{s,R}$. When the admissible frequency deviations of up to $0.4\%$ during normal and up to $5\%$ during critical system states (VDE, 2024a) are neglected, it follows that $\Omega_s := f_s/f_{s,R} \approx 1$. This assumption leads to equal (normalized) power $P = \Omega_s M$ and torque $M$ quantities in Eq. (1a). $P_{m,s}$ and $P_{e,s}$ are defined as the sum of mechanical power $p_m$ and electrical power $p_e$ of all (V)SMs, respectively, both normalized to $p_{s,R}$. More precisely, for a (V)SM, $p_m$ is the mechanical power of the (virtual) turbine, $p_e$ is the electrical power of the (virtual) SM, and $\Theta \neq 0$ is the (virtual) total drivetrain moment of inertia. For grid-following WTs (without VSM), $p_m$ and $p_e$ are the mechanical and electrical WT power, respectively, but the grid-connected moment of inertia is $\Theta = 0$ due to decoupled physical WT inertia. Thus, assuming $p_m \approx p_e$, WTs or all IBRs that are operating under grid-following control can be neglected in Eq. (1a), i. e. only (V)SMs contribute to limiting the initial ROCOF $\dot{\Omega}_s$.

For a SM, the inertia constant $H := E_{kin}/p_R$ is defined as the ratio of the kinetic energy $E_{kin}$ and the rated power $p_R$. Similarly, for a VSM-controlled WT, the virtual inertia constant $H_v := E_{kin,v}/p_R$ is defined as the ratio of the VSM kinetic energy $E_{kin,v}$ and the WT rated power $p_R$. Note that $E_{kin,v}$ differs from the WT physical kinetic energy in general. In particular, a (V)SM always rotates near synchronous speed, whereas the WT speed depends on wind and operating conditions. In contrast to the WT physical inertia constant, the virtual inertia constant $H_v$ is a tunable control parameter. Finally, aggregating all (V)SMs leads to the system inertia $H_s$ in Eq. (1a). However, this is only valid for a proper tuning of $H_v$ because, e. g., emulating a high $H_v$ may not be feasible due to output power limitations. SMs provide an overload capability of 3 to 5 times, whereas IBRs only allow for an overloading of 1 to 1.5 times, which limits the VSM inertial power response depending on the ROCOF (ESIG, 2022). For a VSM-controlled WT, choosing a high $H_v$, e. g. $H_v > H$ with physical WT inertia constant $H$, increases the inertial grid support for low ROCOFs but increases the risk of undesired output power saturation for higher ROCOFs (Höhn et al., 2024). This has to be taken into account when replacing physical inertia by virtual inertia in future power systems.

WT curtailment or derating strategies provide power reserve, e. g. for primary frequency or droop control (Kanev and van de Hoek, 2017; Bossanyi et al., 2020; Clark et al., 2010). For inertia provision, derating based on the maximum rotation strategy (MRS) additionally increases the WT kinetic energy reserve (Ramtharan et al., 2007; Aho et al., 2014; Meseguer Urban et al., 2019; Thommessen and Hackl, 2024). Although derating strategies enhance grid frequency support, they also reduce WT power efficiency. WF and system operators should find a Pareto optimal strategy that considers system stability and efficiency to avoid unnecessary curtailment of renewables.

Consequently, WF inertia forecasting is essential for reliable inertia provision through adequate WT derating and precise tuning of VSM inertia.

**3 Methodology**

The proposed approach combines online and offline calculations, as depicted in the overview of Fig. 1. First, a data-driven weather forecast model predicts the site ambient wind conditions. These ambient conditions serve as input to the WF model,

[Figure]

**Figure 1.** Overview of the proposed WF inertia forecasting approach, which predicts the maximum deliverable inertia constant $H_{v,max}$ of the WF based on online weather forecasting and offline calculated LUTs.

which incorporates the aerodynamic characteristics of all $n$ WTs in the WF, given by the power coefficients $c_p^{[1]}, \ldots, c_p^{[n]}$ and the thrust coefficients $c_t^{[1]}, \ldots, c_t^{[n]}$. The WF model outputs the local wind speeds $v_w^{[1]}, \ldots, v_w^{[n]}$, which are fed back to lookup tables (LUTs) for the power and thrust coefficients. These LUTs of the form $c_p, c_t = f(v_w, P_{set})$ are obtained through offline calculation of the WT steady states $x_0$ for all WT operating points defined by wind speed $v_w$ and power setpoint $P_{set}$ (in % of available power at the MPP). The wake model iteratively computes local wind speeds at all WTs. Additional LUTs of the form $H_{v,max} = f(v_w, P_{set})$ are calculated offline in Fig. 1 by solving optimization problems, which maximize the VSM inertia constant $H_v$ for a given WT operating point $(v_w, P_{set})$ and a reference frequency event defined by grid codes. More precisely,

an optimization algorithm iteratively runs simulations of the WT response to a ROCOF $\dot{\Omega}_s(t)$ with different $H_v$ to find the maximum VSM inertia constant $H_{v,max}$ that the WT can provide without violating operating constraints. With the frequency event starting at $t = t_0$, the operating constraints ensure that the WT states $x(t)$ are within their admissible value range for all $t \geq t_0$. The LUTs $H_{v,max} = f(v_w, P_{set})$ are evaluated online in Fig. 1 to map the WT operating points to the maximum inertia constants $H_{v,max}^{[1]}, \ldots, H_{v,max}^{[n]}$ of all $n$ WTs. Finally, assuming an optimal $H_v = H_{v,max}$ tuning for each WT in the WF and aggregating $H_{v,max}^{[1]}, \ldots, H_{v,max}^{[n]}$, yields the inertia provision in terms of the maximum inertia constant $H_{v,max}$ at WF level. The proposed approach is generic because it is applicable to different modeling and control formulations of WTs and WFs.

Although taking wake effects into account for the initial conditions, the proposed approach assumes that the wind conditions do not change for the duration $\Delta t$ of the frequency event. Despite the volatile nature of real wind profiles, such an assumption for inertia forecasting at the WF level is reasonable, because of an expected averaging effect of any local fluctuations due to the aggregation over several WTs. Moreover, the change in wake behavior during the inertial response is typically probably negligible due to the propagation delay of farm flow effects. For example, for a moderate-sized onshore WT with a rated wind speed of $10\,\mathrm{m\,s^{-1}}$ and a rotor diameter of $130\,\mathrm{m}$, WTs are usually placed apart at a $2\,\mathrm{D}$ to $5\,\mathrm{D}$ distance in an optimal layout design subject to spacing constraints (Stanley et al., 2022). For the worst-case scenario, considering a very short $2\,\mathrm{D}$ spacing, any change in control action on the upstream WT will take ca. $26\,\mathrm{s}$ to reach the downstream WT. This time duration is much greater than the inertial response time or the duration of a severe ROCOF, which lasts only a few seconds.

**3.1 Ambient WF wind forecast**

Wind conditions are forecasted using fully connected neural networks (FCNNs) based upon the methods discussed in Anand et al. (2024). The training targets are the north-aligned component $v_w^u$ and the east-aligned component $v_w^v$ of the wind measurements at the site over the forecast horizon. Features from the two numerical weather prediction (NWP) models ICON-EU and ARPEGE are used as input data (Zängl et al., 2015; P. Courtier et al., 1991). Furthermore, input data also include lag characteristics, i. e., targets for the $u$ and $v$ components of previous timestamps. The choice of forecast horizon may range from several minutes to several hours, depending on the application use case. For example, for a short-term availability prediction, a forecast horizon of a few minutes to one hour is relevant. However, a forecast horizon of up to 36 hours can be necessary for energy market applications.

The probabilistic wind forecast is obtained using a machine learning-based model that utilizes Gaussian mixture distributions formed by superimposing several normal distributions. The resulting probability distribution is given by

$$p(x) = \sum_i^n w_i N(x|\mu_i, \sigma_i), \tag{2}$$

where $w_i$ represents the weight, $\mu_i$ the mean and $\sigma_i$ the standard deviation of the $i$-th Gaussian normal distribution. Due to the long forecast horizon, an ensemble method consisting of several FCNNs was utilized to predict the parameters $w_i$, $\mu_i$, and $\sigma_i$, where each network is trained only on a specific segment of the overall forecast horizon. This approach was chosen due to its ability to deliver an improved forecast accuracy for each individual segment, as opposed to using networks designed to forecast over the entire time horizon. By focusing on shorter segments, the model can better capture dynamic variations and

nuances in the data, leading to more precise predictions. The final forecast is obtained by combining the outputs from multiple ensemble networks, each trained on a specific segment of the data. This ensemble method enhances the overall reliability and accuracy of the forecast. In particular, using a configuration with four networks proved to be an effective compromise, striking a balance between maintaining robustness and minimizing the training time required. This formulation allows for sufficient model flexibility while optimizing computational efficiency, making it a practical choice for operational forecasting.

To reduce the number of input parameters for the FCNNs, a feature-selection algorithm is applied to each of the FCNNs within the ensemble. This is followed by a hyper-parameter optimization process to determine an appropriate number $n$ of normal distributions for the mixed distribution, and to fine-tune both the individual FCNN architectures and the training optimizer. The hyper-parameter optimization is automated and utilizes policy gradients with parameter-based exploration (PGPE) (Sehnke et al., 2010). The training process employs the Adam optimizer, using a mean squared error loss function (Kingma and Ba, 2014). The dataset is divided into training (88 %) and validation (12 %) subsets.

**3.2 Local WT operating point prediction**

An engineering wake model is employed to predict the local inflow conditions at each WT within the WF at steady-state (NREL, 2022). The wake model takes ambient weather forecasts as inputs and models the wake position and velocity deficit within the WF, for given turbine characteristics and operational setpoints. This results in local wind condition forecasts at each WT, which is crucial for accurate performance prediction. More precisely, offline computed WT LUTs of the form $c_{\mathrm{p}}, c_{\mathrm{t}} = f(v_{\mathrm{w}}, P_{\mathrm{set}})$ map local wind speed $v_{\mathrm{w}}$ and power setpoint $P_{\mathrm{set}}$ to power coefficient $c_{\mathrm{p}}$ and thrust coefficient $c_{\mathrm{t}}$. In general, $v_{\mathrm{w}}$ at the downstream WT depends on $c_{\mathrm{t}}$ of the upstream WT. Thus, the wake model iteratively computes the local wind speeds at all WTs, as shown in Fig. 1.

**3.3 Grid-forming capability of WTs**

This section introduces a general approach used for evaluating the grid-forming capability of WTs in terms of maximum inertia provision. First, we present the proposed optimization problem to evaluate maximum deliverable inertia. Then, we develop two solutions of the optimization problem. The first produces a simplified result derived from the formulations in the existing literature. This is followed by a second complete numerical solution, which utilizes a dynamic model of the WT inertial response within the optimization.

**3.3.1 Optimization problem for maximum inertia provision**

The maximum feasible VSM inertia constant $H_{\mathrm{v,max}}$ is obtained by solving the optimization problem

$$\forall t \geq t_0 : H_{\mathrm{v,max}} := \arg\max_{H_{\mathrm{v}}} \{H_{\mathrm{v}}\}, \mathrm{s.\,t.} \left\{ \begin{array}{c} \Omega(t) \geq \Omega_{\mathrm{min}} \\ M_{\mathrm{e}}(t) \leq M_{\mathrm{e,max}} \\ \dot{M}_{\mathrm{e}}(t) \leq \dot{M}_{\mathrm{e,max}} \\ P_{\mathrm{e}}(t) \leq P_{\mathrm{e,max}} \end{array} \right\}. \tag{3}$$

Here, the WT rotor speed $\Omega$ is expressed in per unit (p. u.) of rated WT rotor speed $\omega_R$ (at the low speed shaft in $\mathrm{rad\,s^{-1}}$), the WT electromagnetic torque $M_e$ is in p. u. of rated torque $m_R$ (at the low speed shaft in N m), the WT electromagnetic torque rate $\dot{M}_e := \frac{d}{dt} M_e$ is in $\mathrm{s^{-1}}$, the WT electrical power $P_e$ is in p. u. of rated power $p_R = \omega_R m_R$ (in W), and $\Omega_{min}$, $M_{e,max}$, $\dot{M}_{e,max}$, $P_{e,max}$ denote the corresponding limits. Note that, although the objective function and optimization argument in Eq. (3) are the same, solving Eq. (3) is not trivial due to the nonlinear optimization constraints. Depending on the grid codes and WT design, Eq. (3) may include additional constraints, e. g. for recovery power limits (see Appendix G) or for stall limits, which are implicitly taken into account here by $\Omega_{min}$ in Eq. (3).

**3.3.2  Simplified solution**

Lee et al. (2016) evaluate the capability of grid-following WTs to emulate inertia by considering WT rotor speed or available kinetic energy reserve. Here, unlike Lee et al. (2016), we derive a simplified solution of Eq. (3) for grid-forming WTs that takes all operating constraints into account, and not only the WT rotor speed limits.

Neglecting any changes of aerodynamic conditions during the inertial response, i. e. assuming constant wind speed $v_w$, blade pitch angle $\beta$ and tip speed ratio $\lambda := \Omega \omega_R r / v_w$ with WT radius $r$, it follows that the WT aerodynamic or mechanical power $P_m$ (in p. u. of $p_R$) is constant, i. e.

$$\forall t_0 \le t \le t_0 + \Delta t : \left\{ \begin{array}{l} v_w(t) = v_w(t_0) =: v_{w,0} \\ \beta(t) = \beta(t_0) =: \beta_0 \\ \lambda(t) = \lambda(t_0) =: \lambda_0 \end{array} \right\} \qquad \Rightarrow \qquad P_m(t) = P_m(t_0) =: P_{m,0}. \tag{4}$$

For the simplified solution, we assume that the ROCOF $\dot{f}_s$ is constant and equal to the worst-case initial ROCOF, until reaching the minimum frequency nadir $f_{s,min}$ at time $t = t_{s,min}$, i. e. the considered time duration is given by

$$\Delta t := t_{s,min} - t_0 = \frac{f_{s,min} - f_{s,R}}{\dot{f}_s}. \tag{5}$$

Assuming that the initial electrical power equals the mechanical power in Eq. (4), i. e. $P_{e,0} = P_{m,0}$, and approximating the electrical power change during $\Delta t$ by an ideal power pulse $\Delta P_e$ according to the simplified inertial response in Eq. (1a), the electrical power constraint in Eq. (3) simplifies to

$$P_e := P_{e,0} + \Delta P_e := P_{m,0} + 2 H_v \dot{\Omega}_{s,max} \le P_{e,max}, \tag{6}$$

where the normalized worst-case ROCOF magnitude is $\dot{\Omega}_{s,max} := |\dot{f}_{s,0}| / f_{s,R} > 0$. It follows that additional output power $\Delta P_e := 2 H_v \dot{\Omega}_{s,max}$ is extracted from the WT kinetic energy reserve, and the minimum rotor speed constraint in Eq. (3)

simplifies to
$$\int_{t_0}^{t_{s,min}} (P_m - P_e)\, dt = E_{kin}(t_{s,min}) - E_{kin}(t_0),$$

from which we get
$$\Delta P_e \Delta t = H \left( \Omega_0^2 - \Omega(t_{s,min})^2 \right),$$

and finally
$$\Omega(t_{s,min}) = \sqrt{\Omega_0^2 - 2 \frac{H_v}{H} (1 - \Omega_{s,min})} \ge \Omega_{min}, \tag{7}$$

where $\Omega_0 := \Omega(t_0)$ and the physical total WT drivetrain inertia constant is $H$. Note that Eq. (7) depends on the normalized frequency nadir $\Omega_{s,min} := f_{s,min}/f_{s,R}$ and not explicitly on the ROCOF, which justifies the aforementioned assumption of a constant ROCOF in Eq. (5). Based on Eq. (6) and Eq. (7), the torque constraint in Eq. (3) simplifies to

$$\max M_e = \frac{P_e}{\Omega(t_{s,min})} \leq M_{e,max}. \tag{8}$$

Finally, with Eqs. (6 – 8) and the simplified torque rate constraint derived in the appendix Eq. (A5), a nonlinear optimization algorithm (MATLAB, 2025) solves Eq. (3) for given initial values $\Omega_0$ and $P_{m,0}$.

**3.3.3 Complete numerical solution**

The optimization problem expressed by Eq. (3) can be solved in a more general way, where dynamic simulations of the WT inertial response to a worst-case or reference frequency event replace the aforementioned simplified expressions. More precisely, an optimization algorithm iterates the simulations with varying $H_v$ to find the maximum inertia constant $H_{v,max}$ that does not violate any operating limits (see Fig. 1). Clearly, this approach is generic due to its applicability to different WT models and their controllers. Moreover, this approach allows for more accurate solutions. For instance, derating strategies can provide additional wind power reserves (Kanev and van de Hoek, 2017; Meseguer Urban et al., 2019; Thommessen and Hackl, 2024), which are only taken into account by the complete numerical solution but not by the simplified one. For the iterative simulations during optimization, we rely on appropriate WT modeling, with steady-state initialization derived in Appendix B.

Inertia provision requires power headroom. Accordingly, all saturations or manipulations of the WT power reference $P_{ref}$ for protection are not just removed in the WT control model, but are converted into corresponding inequality constraints, i. e.

$$\forall t \geq t_0 : \boldsymbol{c} := \begin{pmatrix} c_1 \\ c_2 \\ c_3 \\ c_4 \end{pmatrix} := \begin{pmatrix} \Omega_{min} - \min \Omega(t), \\ \max M_e(t) - M_{e,max}, \\ \max \dot{M}_e(t) - \dot{M}_{e,max}, \\ \max P_e(t) - P_{e,max} \end{pmatrix}, \quad \text{where } \forall i \in \{1,2,3,4\} : c_i \geq 0. \tag{9}$$

Based on Eq. (9), a nonlinear optimization algorithm (MATLAB, 2025) solves the optimization problem in Eq. (3). The $i$-th constraint is considered active if $c_i = 0$, or inactive if $c_i > 0$.

**3.4 WT modeling and control**

Figure 2 depicts the overall WT modeling and control used in this work. During normal operation, the MRS-based power setpoint $\overline{P}_{set}^{\star}$ defines the electromagnetic torque $M_e$. During frequency events, $M_e$ additionally depends on the active power droop and inertia provision through VSM control. Details of these torque controllers follow in Sects. 3.4.2 – 3.4.5.

The WT modeling approach utilized in this work is based on the reference design in Bortolotti et al. (2019), except for the pitch controller that here does not include a tip speed constraint below rated wind speed. Thus, at the rated wind speed $v_{w,R} = 9.8\,\mathrm{m\,s^{-1}}$, the WT operates at its MPP with optimal tip speed ratio $\lambda^{\star} = 8.5$. Consequently, the rated tip speed $\omega_R r = \lambda^{\star} v_{w,R} = 83.3\,\mathrm{m\,s^{-1}}$ (slightly) exceeds the tip speed limit of $80\,\mathrm{m\,s^{-1}}$ assumed in Bortolotti et al. (2019). The resulting rated

[Figure]

**Figure 2.** WT modeling and control for solving the optimization problem in Eq. (3). The simplified control representation for the iterative simulations during optimization is derived based on the VSM control for WTs in Thommessen and Hackl (2024). All saturations or manipulations of the power reference $P_{\text{ref}}$ for WT protection have been removed and converted into corresponding optimization constraints.

WT rotor speed $\omega_R$ is ca. 4.2 % larger than the rated value in Bortolotti et al. (2019), but still ca. 5.3 % smaller than the maximum assumed rotor speed limit. The higher speed rating increases not only the rated power but also the rated kinetic energy reserve for inertia provision compared with Bortolotti et al. (2019). Neglecting for simplicity any conversion losses from mechanical to electrical power, the WT physical inertia constant at rated speed becomes

$$H_R = \frac{\frac{1}{2}\Theta\omega_R{}^2}{p_R} = 3.26\,\text{s}. \tag{10}$$

The WT physical inertia constant $H$ (which should be better called "inertia variable" due to the variable $\Omega$) is proportional to the WT kinetic energy $E_{kin}$, i. e.

$$H := \frac{E_{kin}}{p_R} := \frac{\frac{1}{2}\Theta\omega^2}{p_R} = \Omega^2 H_R. \tag{11}$$

Note that, for (directly grid-connected) SMs (of conventional power plants), it follows that $H = \Omega_s{}^2 H_R \approx H_R$ due to $\Omega_s \approx 1$.

The blade pitch angle system modeling and control is based on Thommessen and Hackl (2024, Sect. II.C and Sect. III.A). The main objective of the pitch control is to increase the pitch angle reference $\beta_{ref}$ for above-rated wind speeds to limit the rotor speed to $\Omega = 1$. In addition to $\Omega$, the wind speed $v_w$ is also required as input for the pitch control in Fig. 2, in order to adapt the lower pitch angle limit based on the tip speed ratio, i. e. $\beta_{ref} \geq \beta_{min}(\lambda)$. This is more relevant for derating than for MPPT.

**3.4.1 Aeroelastic and mechanical model**

The power coefficient $c_p$ and the thrust coefficient $c_t$ are modeled as functions of tip speed ratio $\lambda$ and blade pitch angle $\beta$ by the corresponding LUTs, see Fig. 3. The fore-aft deflection of the WT tower, excited by the thrust force $F_t$, is modeled as a mass-spring-damper oscillator with mass $m_t$, damping coefficient $d_t$ and stiffness coefficient $k_t$. The aeroelastic model outputs the WT mechanical torque $M_m$ (in p. u. of $m_R$) for given inputs $(v_w, \beta, \Omega)$, see also Fig. 2, i. e.

$$P_w = F_w v_w / p_R = \tfrac{1}{2}\rho\pi r^2 v_w{}^3 / p_R, \tag{12a}$$

$$M_m = \frac{P_m}{\Omega} = \frac{P_w c_p(\lambda,\beta)}{\Omega}, \qquad \lambda = \Omega\frac{\omega_R r}{\widetilde{v}_w}, \qquad \widetilde{v}_w = v_w - \dot{s}_t, \tag{12b}$$

$$\ddot{s}_t = \frac{1}{m_t}\left(F_t - d_t\dot{s}_t - k_t s_t\right), \qquad F_t = F_w c_t(\lambda,\beta), \quad s_{t,0} = \frac{F_w c_t(\lambda_0,\beta_0)}{k_t}, \quad \dot{s}_{t,0} = 0, \tag{12c}$$

with wind power $P_w$ (in p. u. of $p_R$), wind-generated force $F_w$, air density $\rho$, relative wind speed $\widetilde{v}_w$, WT fore-aft tower displacement $s_t$, and initial steady-state values $s_{t,0}, \dot{s}_{t,0}, \lambda_0, \beta_0$. Indicating with $H_R$ (in s) the WT total physical drivetrain inertia constant, the WT mechanical dynamics are approximated by a one-mass model, i. e. (Thommessen and Hackl, 2024)

$$\dot{\Omega} = \frac{M_m - M_e}{2H_R}, \qquad \Omega(0) = \Omega_0. \tag{13}$$

[Figure]

**Figure 3.** Power coefficient $c_p$ **(a)** and thrust coefficient $c_t$ **(b)**, as functions of tip speed ratio $\lambda$ and blade pitch angle $\beta$. The MPP is indicated with the symbol ★. $(\lambda^\star, \beta^\star) = (8.5, 1.1^\circ)$, $c_p^\star = c_p(\lambda^\star, \beta^\star) = 0.48$, $c_t^\star = c_t(\lambda^\star, \beta^\star) = 0.96$.

**3.4.2 Maximum rotation strategy**

With the MPPT torque curve $M_{\mathrm{mppt}} := M_{\mathrm{mppt}}(\Omega)$ shown in Fig. 4, the electrical power for MPPT is given by

$$P_{\mathrm{mppt}} := \Omega M_{\mathrm{mppt}}. \tag{14}$$

Below rated wind speed in region II, $M_{\mathrm{mppt}}$ increases proportionally to $\Omega^2$ for optimal operation at the MPP. Above rated wind speed in region III, the pitch control limits the WT rotor speed to $\Omega = 1$ such that $P_{\mathrm{mppt}} = M_{\mathrm{mppt}} = 1$. Below cut-in wind speed $v_{\mathrm{w,cut\text{-}in}}$ in region I, the WT does not generate power, i.e. $M_{\mathrm{mppt}} = 0$ for $\Omega < \Omega_{\min}$. For a smooth transition to region II, a non-optimal operation is accepted in the small transition region I-II defined by $\Omega_{\min} \leq \Omega \leq \Omega_{\min} + \Delta\Omega_{\mathrm{I\text{-}II}}$, i.e. $M_{\mathrm{mppt}}$ is obtained by multiplying the optimal torque at the MPP by a factor that is linearly interpolated between 0 at $\Omega_{\min}$ and 1 at $\Omega_{\min} + \Delta\Omega_{\mathrm{I\text{-}II}}$. For a more complete description of the MPPT curve, see Thommessen and Hackl (2024, Sect. III.B.1).

[Figure]

**Figure 4.** MPPT torque as a function of rotor speed $\Omega$.

The MRS-based derating maximizes the WT kinetic energy reserve for inertia provision. The derating power setpoint $P_\text{set}$ is defined relative to the MPP, i.e. $P_\text{set} = 1$ corresponds to MPPT and $P_\text{set} < 1$ corresponds to derating. Increasing derating (decreasing $P_\text{set}$) reduces the electrical power setpoint $P_\text{set}^\star := \overline{P}^\star P_\text{set}$ with available power $\overline{P}^\star \in (0,1]$, i.e. the WT accelerates. The tip speed ratio $\lambda$ increases such that the power coefficient $c_\text{p}$ decreases (see Fig. 3). The pitch controller additionally increases $\beta$ if necessary to limit the rotor speed to $\Omega = 1$. In general, the MRS prioritizes increasing WT speed over pitching to provide power reserve. With the measured or estimated rotor-effective wind speed $v_\text{w}$ (Soltani et al., 2013) and the MPPT power coefficient function or LUT $\overline{c}_\text{p}^\star(v_\text{w})$ (see Appendix B4) the available power in Fig. 2 is defined as

$$\overline{P}^\star := P_\text{w}\overline{c}_\text{p}^\star(v_\text{w}) = \tfrac{1}{2}\rho\pi r^2 v_\text{w}{}^3 \overline{c}_\text{p}^\star(v_\text{w})/p_\text{R}. \tag{15}$$

Limiting $P_\text{set}^\star$ by $P_\text{mppt}$ for rotor speed transients and wind measurement errors, the saturated power setpoint is given by [3]

$$\overline{P}_\text{set}^\star := \left\{ \begin{array}{ll} \min(P_\text{set}^\star, P_\text{mppt}), & \text{if } P_\text{set} < 1 \text{ (derating)}, \\ P_\text{mppt}, & \text{if } P_\text{set} = 1 \text{ (MPPT)}. \end{array} \right\} \quad \text{with } P_\text{set}^\star := \overline{P}^\star P_\text{set}. \tag{16}$$

**3.4.3 VSM control**

Grid-forming control is required to limit the initial ROCOF (ESIG, 2022; VDE, 2024a). This paper simplifies the grid-forming VSM control proposed in Thommessen and Hackl (2024), by neglecting fast electromagnetic transients and low-level current control loops. However, the grid synchronization dynamics of grid-forming control define the inertial response and must, therefore, be taken into account.

[Figure]

**Figure 5.** Grid synchronization loop of a freely spinning VSM.

For VSM control, the grid synchronization dynamics are similar to the dynamics of a real (grid-connected) SM, as illustrated in Fig. 5. The VSM acceleration $\dot{\Omega}_\text{v}$ is proportional to the sum of virtual torques, i.e. the VSM mechanical model is based on a
* * *
[3]In Eq. (16), $P_\text{set}^\star$ is ignored for $P_\text{set} = 1$ (MPPT), since (i) no wind measurements are required, and (ii) smaller transient rotor speed overshoots occur due to higher power setpoint adaption. For example, if $\Omega > 1$ due to a wind gust, it follows that $\overline{P}_\text{set}^\star = P_\text{mppt} = \Omega M_\text{mppt} > 1$ whereas $P_\text{set}^\star \leq 1$.

one-mass model with virtual inertia $H_\mathrm{v}$ instead of physical inertia $H_\mathrm{R}$ in Eq. (13). At steady state, the difference between VSM mechanical and electromagnetic torque is zero, i.e. $M_\mathrm{m,v} - M_\mathrm{e,v} = 0$, and the VSM damping torque is zero, i.e. $M_\mathrm{dp,v} = 0$. Since only the inertial response to ROCOFs or electromagnetic changes are of interest, the VSM mechanical torque is set to zero, i.e. $M_\mathrm{m,v} = 0$, resulting in the freely spinning VSM in Fig. 5. Denormalization of the power system frequency, i.e.

330 $\omega_\mathrm{s} = \Omega_\mathrm{s}\omega_\mathrm{s,R}$ with $\omega_\mathrm{s,R} := 2\pi f_\mathrm{s,R}$, and subsequent integration of $\omega_\mathrm{s}$ yields the grid or system angle $\phi_\mathrm{s}$. Similarly, denormalization of the VSM rotor speed, i.e. $\omega_\mathrm{v} = \Omega_\mathrm{v}\omega_\mathrm{s,R}$, and subsequent integration of $\omega_\mathrm{v}$ yields the VSM rotor angle $\phi_\mathrm{v}$. The VSM electromagnetic torque $M_\mathrm{e,v}$ depends on the (real) load angle $\delta := \phi_\mathrm{v} - \phi_\mathrm{s}$ multiplied by the electromagnetic feedback gain $k_\mathrm{e}$. Due to unknown $\phi_\mathrm{s}$ or $k_\mathrm{e}$, the VSM controller calculates the torque or power feedback directly based on current and grid voltage measurements. The VSM damping torque $M_\mathrm{dp,v}$ is proportional to the VSM slip $\Omega_\mathrm{v} - \Omega_\mathrm{s}$, which emulates the effect of

335 damper windings in SMs. However, unlike real SMs, the VSM enables flexible tuning of the VSM damping $D_\mathrm{v}$. Grid voltage measurements are required to determine $\Omega_\mathrm{s}$ for VSM damping torque calculation.

The VSM power for inertia provision, added to the power setpoint $\overline{P}_\mathrm{set}^{\star}$ in Fig. 2, is defined as

$$P_\mathrm{v} = \Omega_\mathrm{v} M_\mathrm{e,v} \tag{17}$$

where $\Omega_\mathrm{v}$ and $M_\mathrm{e,v}$ are given by the grid synchronization loop in Fig. 5 with input $\dot{\Omega}_\mathrm{s}$.

340 The inertial response in the Laplace domain of the VSM is given by (Thommessen and Hackl, 2024)

$$\frac{M_\mathrm{e,v}}{\dot{\Omega}_\mathrm{s}} = \frac{-k_\mathrm{e}\omega_\mathrm{s,R}}{s^2 + 2\zeta_\mathrm{v}\omega_\mathrm{n,v}s + \omega_\mathrm{n,v}^2}, \quad \omega_\mathrm{n,v}^2 := \frac{k_\mathrm{e}\omega_\mathrm{s,R}}{2H_\mathrm{v}}, \quad \zeta_\mathrm{v} := \frac{D_\mathrm{v}}{\sqrt{8H_\mathrm{v}k_\mathrm{e}\omega_\mathrm{s,R}}} = 1 \;\Rightarrow\; D_\mathrm{v} := D_\mathrm{v}(H_\mathrm{v}) := \sqrt{8H_\mathrm{v}k_\mathrm{e}\omega_\mathrm{s,R}} \tag{18}$$

with natural angular velocity $\omega_\mathrm{n,v}$ and damping ratio chosen as $\zeta_\mathrm{v} = 1$ to avoid overshooting. With grid synchronized VSM speed $\Omega_\mathrm{v} = \Omega_\mathrm{s} \approx 1$ in Eq. (17) and setting $s = 0$ in Eq. (18), the steady-state VSM power for a constant ROCOF simplifies to

$$P_\mathrm{v} = \Omega_\mathrm{v} M_\mathrm{e,v} \approx M_\mathrm{e,v} \approx -2H_\mathrm{v}\dot{\Omega}_\mathrm{s}. \tag{19}$$

345 The power system dynamics in Fig. 5 are defined by a reference frequency event. The electromagnetic feedback in Fig. 5 depends on the load angle given by the angle difference between VSM and grid, i.e. $M_\mathrm{e,v} = k_\mathrm{e}\delta = k_\mathrm{e}(\phi_\mathrm{v} - \phi_\mathrm{s})$. For simplicity, this paper assumes a constant electromagnetic feedback gain $k_\mathrm{e}$. Actually, $k_\mathrm{e}$ depends on the WT operating point and the WT grid connection, i.e. $k_\mathrm{e}$ is a nonlinear function of the load angle delta $\delta$, the grid voltage and the grid impedance (VDE, 2024a; Ghimire et al., 2024). Type 3 WTs use doubly-fed induction machines (DFIMs), where $k_\mathrm{e}$ also depends on the DFIM rotor

350 current or excitation level (Thommessen and Hackl, 2024, Sect. III.E). If not negligible, the dependency of $k_\mathrm{e}$ on $\delta$ and the excitation level should be taken into account based on the WT operating point. With admissible limits for grid voltage and impedance defined by grid codes (VDE, 2024a), $k_\mathrm{e}$ should be chosen based on the WT grid connection, see Appendix C.

This paper assumes internal damping of the VSM (Roscoe et al., 2020; Thommessen and Hackl, 2024), i.e. the damping torque $M_\mathrm{dp,v}$ in Fig. 5 is solely virtual and is not converted into real electrical output power. In contrast, for an external damping

355 of a real SM, the damping power is part of the electrical output power (Roscoe et al., 2020). In this regard, $H_\mathrm{v}$ of a VSM and $H$ of a (real) SM differ, i.e., assuming $H_\mathrm{v} = H$ and equal damping gains, the actually extracted kinetic energy during the inertial response is smaller for a VSM-controlled WT than for a (real) SM, see Fig. 6. The damping energy corresponds to the area

between the two curves in Fig. 6. Strictly speaking, the VSM concept violates the law of conservation of energy since the VSM braking energy is not fully converted into electrical energy. However, high internal damping avoids power overshoots and is beneficial for grid frequency stability (Roscoe et al., 2020; Thommessen and Hackl, 2024). Also, grid codes (VDE, 2024a) require sufficient damping and consider (internal) damping power separately from electrical output power, see VDE (2024a, Kap. 5.1.1.11, Anmerkung 1). Finally, $H_\mathrm{v}$ is comparable to $H$ of a real SM when neglecting the transient damping, i. e. when considering the quasi-steady-state power change $\Delta P = -2H_\mathrm{v}\dot{\Omega}_\mathrm{s}$, as shown in Fig. 6. Thus, $H_\mathrm{v}$ is a suitable measure for the inertial power response and the grid frequency support by inertia provision. [4]

[Figure]

**Figure 6.** Inertial power response to a ROCOF of $\dot{\Omega}_\mathrm{s} = -4\,\%\,\mathrm{s}^{-1}$ for $H_\mathrm{v} = 3\,\mathrm{s}$ with quasi-steady-state amplitude $\Delta P$ approximated by Eq. (19). The VSM achieves the desired internal damping of the VSM output power, whereas the VSM braking power overshoots. The VSM output power equals the WT electrical power change. The VSM braking power equals the electrical power of an equivalent real SM with external damping, i. e. the SM braking energy is fully converted into electrical energy according to the law of conservation of energy.

This paper assumes an ideal inertial power response, i. e. the VSM power $P_\mathrm{v}$ is added to the original machine power setpoint $\overline{P}_\mathrm{set}^\star$ in Fig. 2. Moreover, the final electromagnetic torque equals the electromagnetic torque reference, i. e. $M_\mathrm{e} = M_\mathrm{e,ref}$ in Fig. 2, neglecting low-level current controls with closed-loop time constants that are significantly smaller than the ones of high-level WT or VSM control. Clearly, this is a simplified representation of the actual VSM control, which adjusts the voltage or current phase angle based on the VSM angle $\phi_\mathrm{v}$ to achieve the grid-forming capability (ESIG, 2022; Thommessen and Hackl, 2024). Although the implementation details are beyond the scope of this paper, the simplified representation should take into account the general differences between existing VSM control strategies, as discussed in Appendices D – F.
* * *
[4]The recent draft VDE (2024b) for certification of grid-forming IBRs quantifies inertia provision by the mean power change over a time window starting $0.5\,\mathrm{s}$ after the ROCOF change and ending at the beginning of the next ROCOF change during the reference frequency event, i. e. $T_\mathrm{A} := \mathrm{mean}\,|\Delta P(t)/\dot{\Omega}_\mathrm{s}| \approx 2H_\mathrm{v}$. Due to a constant initial ROCOF for $1\,\mathrm{s}$ during the reference frequency event (see $\Omega_\mathrm{s}$ in Fig. 10), the considered time window for quantifying inertia provision for the initial ROCOF would be $0.5\,\mathrm{s} \leq t \leq 1\,\mathrm{s}$. For simplicity, this paper quantifies inertia provision by the control parameter $H_\mathrm{v}$ (see also Fig. 6), which results in a (slight) overestimation of the inertia provision compared to VDE (2024b).

**3.4.4 MPPT compensation**

For a negative ROCOF, the WT output power increases for inertia provision, which decelerates the WT, i. e. $\Omega$ decreases. The MPPT would counteract the deceleration or the desired inertial response by reducing $P_{\text{mppt}}$ for decreasing $\Omega$ according to Eq. (14). To avoid this, the so-called MPPT compensation manipulates the MPPT input $\Omega$ (Duckwitz, 2019; Thommessen and Hackl, 2024). This paper simplifies the MPPT compensation proposed in Thommessen and Hackl (2024, Sect. III.B.2). The speed change due to inertia provision is estimated by replacing the numerator of the one-mass model in Eq. (13) by the inertial torque change $P_{\text{v}}/\Omega$. Thus, the manipulated MPPT input, equal to the theoretical WT rotor speed for zero inertia provision, is given by

$$\Omega_{H0} := \Omega + \left\{ \begin{array}{ll} \min\left\{ a, \ \dfrac{1}{2H_{\text{R}}} \displaystyle\int_{t_0}^{t} \dfrac{P_{\text{v}}}{\Omega} \mathrm{d}t \right\}, & \text{if } |\dot{\Omega}_{\text{v}}| > \epsilon \ (\text{active MPPT compensation}), \\[2ex] 0, & \text{if } |\dot{\Omega}_{\text{v}}| \leq \epsilon \ (\text{inactive MPPT compensation}), \end{array} \right\}, \quad a := \max\{1 - \Omega, 0\}, \quad (20)$$

with the threshold $\epsilon$ used for detecting active inertia provision based on the VSM acceleration $\dot{\Omega}_{\text{v}} \approx \dot{\Omega}_{\text{s}}$. The integral in Eq. (20) is reset to zero for inactive MPPT compensation. The actual implementation of Eq. (20) includes an additional rate limiter, which ensures $|\dot{\Omega}_{H0}| \leq \dot{\Omega}_{H0,\text{max}}$ with maximum acceleration $\dot{\Omega}_{H0,\text{max}}$ for a smooth transition between active and inactive MPPT compensation (Thommessen and Hackl, 2024, Sect. III.B.2).

Assuming active MPPT compensation, a prolonged MPP deviation during a long time period with a small negative ROCOF would lead to excessive WT rotor deceleration. Thus, the threshold $\epsilon$ in Eq. (20) should not be chosen too small. This also implies less inertia provision for small negative ROCOFs $|\dot{\Omega}_{\text{s}}| \approx |\dot{\Omega}_{\text{v}}| \leq \epsilon$ than expected by $H_{\text{v}}$, due to inactive MPPT compensation. Also, for $\epsilon < |\dot{\Omega}_{\text{s}}| < \dot{\Omega}_{\text{s,max}}$, there may be cases where the inertia provision is (slightly) smaller than expected by $H_{\text{v}}$ if output power saturation is required to protect the rotor speed due to prolonged MPP deviations. However, the proposed approach ensures unsaturated or full inertia provision when reaching the worst-case or reference ROCOF $|\dot{\Omega}_{\text{s}}| = \dot{\Omega}_{\text{s,max}}$.

**3.4.5 Active power droop control**

In addition to inertia provision, which supports grid frequency by injecting inertial VSM power $P_{\text{v}}$ proportional to the ROCOF, active power droop control supports grid frequency by injecting (saturated) droop power $\overline{P}_{\text{d}}$ proportional to the frequency deviation $\Delta\Omega_{\text{s}} := 1 - \Omega_{\text{s}}$. Thus, the final power reference in Fig. 2 is given by $P_{\text{ref}} = \overline{P}_{\text{set}}^{\star} + P_{\text{v}} + \overline{P}_{\text{d}}$. For WTs, droop control is inactive during normal operation within a tolerance band of $|\Delta\Omega_{\text{s}}| \leq 0.4\,\%$ (VDE, 2024a), i. e. the (unsaturated) droop power is $P_{\text{d}} = \overline{P}_{\text{d}} = 0$ in Fig. 2. During a critical system state outside of the tolerance band, the WTs have to support grid frequency by a proportional power adaptation when possible. This means that $\overline{P}_{\text{d}} = P_{\text{d}} > 0$ is only required if wind power reserves are available due to previous derating (VDE, 2024a). For our control strategy, this implies that the inertia provision based on additional kinetic energy extraction is prioritized over droop control; see Appendix I for details.

Ignoring the two max-blocks in Fig. 2, the maximum droop power $P_{\text{d,max}}$ is given by the total currently available power $\min(\overline{P}^{\star}, P_{\text{mppt}})$ minus the sum of the power setpoint $\overline{P}_{\text{set}}^{\star}$ and the VSM power $P_{\text{v}}$. The saturation $\overline{P}_{\text{d}} := \min(P_{\text{d}}, P_{\text{d,max}})$ prevents excessive WT overloading since, without it, the droop power $P_{\text{d}}$ would add to the inertial power even if the output or

reference power $P_{\mathrm{ref}}$ already exceeds the available one. In other words, the WT control prioritizes inertia provision over droop control. Similarly, for real (grid-connected) SMs, the droop control or speed governor response time is significantly slower than the SM inertial response, i. e. only the SM inertial power limits the initial ROCOF.

The additional saturations by the two max-blocks in Fig. 2 ensure that the droop control power $P_{\mathrm{d}}$ does not counteract the VSM power $P_{\mathrm{v}}$ for inertia provision. Without the upper max-block, $P_{\mathrm{d}} = P_{\mathrm{d,max}} < 0$ could counteract $P_{\mathrm{v}} > 0$ for a high negative initial ROCOF; without the lower max-block, $P_{\mathrm{d}} = P_{\mathrm{d,max}} > 0$ could counteract $P_{\mathrm{v}} < 0$ more than expected for a subsequent positive ROCOF during frequency recovery. The presented control is a simplified version of the actual control with dynamic droop saturation of Thommessen and Hackl (2024, Sect. III.F).

**4 Results**

This section presents results regarding different aspects of the proposed WF inertia forecasting approach. First, Sect. 4.1 discusses the WT steady states for different MRS-based deratings (refer to Sect. 3.4.2). Then, Sect. 4.2 demonstrates the simulated WT inertial response to a reference frequency event defined by grid codes, and discusses the mapping of WT operating points to the provision of deliverable inertia. Finally, the overall performance of the proposed WF inertia forecasting is evaluated and compared to the existing approaches in Sect. 4.3.

[Figure]

**Figure 7.** Wind farm layout and wake interactions for ambient wind speed $v_{\mathrm{w}} = 11\,\mathrm{m\,s^{-1}}$ and wind direction $\Gamma_{\mathrm{w}} = 270\,^{\circ}$.

The WF considered here consists of twelve reference WTs from Bortolotti et al. (2019) with operating limits $\Omega_{\mathrm{min}} = 30.81\,\%$, $P_{\mathrm{e,max}} = 105\,\%$, $M_{\mathrm{e,max}} = 106\,\%$, $\dot{M}_{\mathrm{e,max}} = 150\,\%\,\mathrm{s^{-1}}$. The power limit $P_{\mathrm{e,max}}$ is chosen based on the inverter design (Höhn et al., 2024), whereas the other limits are chosen based on the aeroelastic design (Bortolotti et al., 2019). The

420 WTs are arranged in an irregular WF layout on semi-complex terrain characterized by gently rolling hills. Figure 7 shows the WF layout as well as the resulting wake interactions among the WTs in exemplary wind conditions.

Historical data consisting of 2 years of site-specific weather condition measurements are used to train the data-driven weather forecast model. A deterministic model and a probabilistic model predict the weather conditions with a 15-minute resolution for the hour ahead. The deterministic model outputs the expected wind conditions for WF inertia forecasting, whereas, the
425 probabilistic model additionally considers the wind condition uncertainties, enabling uncertainty quantification of the predicted WF inertia.

**4.1 WT steady states**

The WT steady states depend on the WT operating point $(v_\mathrm{w}, P_\mathrm{set})$, defined by wind speed $v_\mathrm{w}$ and power setpoint $P_\mathrm{set}$. Although the WT steady states generally depend on the two dimensions $(v_\mathrm{w}, P_\mathrm{set})$, they are calculated by solving one-dimensional
430 optimization (sub)problems. This is obtained through a case analysis of active operating constraints, see Appendix B. This enables the fast initialization of the WT dynamic model without running time-consuming simulations until reaching steady state. Figure 8 illustrates the WT steady-state conditions as a function of $v_\mathrm{w}$, with $P_\mathrm{set}$ ranging from the minimum considered value of $90\,\%$ (dark blue line) in increments of $1\,\%$ up to the maximum value of $100\,\%$ (dark red line). Note that $P_\mathrm{set} = 1$ corresponds to MPPT, and $P_\mathrm{set} < 1$ corresponds to MRS-based derating. In Fig. 8 (and in all following figures), all normalized quantities
435 (indicated by [%]) are in per unit of rated values, e.g. $F_\mathrm{t,pu} := F_\mathrm{t}/F_\mathrm{t,R}$ with thrust force $F_\mathrm{t} = F_\mathrm{t,R}$ at the rated WT operating point $(v_\mathrm{w} = v_\mathrm{w,R} = 9.8\,\mathrm{m\,s^{-1}}, P_\mathrm{set} = 1)$. The only exception is the power setpoint $P_\mathrm{set}$ defined in per unit of available power $\overline{P}^\star$, see Eq. (16).

In Fig. 8, higher derating or a lower $P_\mathrm{set}$ increases the WT rotor speed $\Omega$ at low wind speeds, e.g. at $v_\mathrm{w} = 5\,\mathrm{m\,s^{-1}}$, such that the kinetic energy or physical inertia constant $H$ increases proportional to $\Omega^2$. For $P_\mathrm{set} = 1$ (MPPT), the blade pitch angle
440 equals its optimal value for below-rated wind speeds in region II, i.e. $\beta = \beta^\star = 1.1\,°$ (see also Fig. 3); on the other hand, for above-rated wind speeds in region III, $\beta$ increases to limit the WT rotor speed to $\Omega = 1$. In addition to $\Omega$, the pitch control requires $v_\mathrm{w}$ as input (see Fig. 2) to adjust the lower pitch angle limit as a function of the tip speed ratio $\lambda = \Omega\omega_\mathrm{R} r/v_\mathrm{w}$, i.e. $\beta_\mathrm{ref} \geq \beta_\mathrm{min}(\lambda) := \arg\max c_\mathrm{p}(\lambda, \beta)$. Considering the plots in the third row of Fig. 8, this $\beta_\mathrm{min}$-adjustment is only relevant for $\lambda > \lambda^\star = 8.5$ due to constant $\beta_\mathrm{min} = \beta^\star$ elsewhere. More precisely, for $P_\mathrm{set} = 1$ (MPPT), the $\beta_\mathrm{min}$-adjustment is only relevant
445 in the small transition region I-II near $v_\mathrm{w,cut-in} = 3.02\,\mathrm{m\,s^{-1}}$; however, for $P_\mathrm{set} < 1$ (derating), the $\beta_\mathrm{min}$-adjustment is also relevant in region II, as the increased tip speed ratio $\lambda > \lambda^\star$ leads to a higher blade pitch angle $\beta = \beta_\mathrm{min}(\lambda) \geq \beta^\star$.

In Fig. 8, after reaching rated rotor speed $\Omega = 1$, the tip speed ratio decreases with increasing wind speed, i.e. $\lambda \propto v_\mathrm{w}^{-1}$, and the blade pitch angle $\beta$ increases to limit the rotor speed to $\Omega = 1$. Both decreasing $\lambda$ and increasing $\beta$ reduce the thrust coefficient $c_\mathrm{t}$ (at least near the optimal operating point, see also Fig. 3). Note that, for MRS-based derating ($P_\mathrm{set} < 1$), the rotor
450 speed reaches $\Omega = 1$ at below-rated wind speeds $v_\mathrm{w} < v_\mathrm{w,R}$. For $\Omega < 1$ and constant $P_\mathrm{set}$, the thrust coefficient $c_\mathrm{t}$ is constant due to constant $\lambda$ and $\beta$. For $\Omega < 1$ and varying $P_\mathrm{set}$, e.g. at $v_\mathrm{w} = 5\,\mathrm{m\,s^{-1}}$, higher derating increases $\lambda$ but only slightly increases $\beta = \beta_\mathrm{min}(\lambda)$ such that $c_\mathrm{t}$ (slightly) increases. This (slightly) increases the thrust force $F_\mathrm{t,pu}$, although the changes are negligible. In contrast, for $\Omega = 1$, higher derating significantly reduces the thrust force due to increasing $\beta$ but constant $\lambda$.

[Figure]

**Figure 8.** WT steady-state conditions as functions of wind speed $v_\mathrm{w}$ for MRS-based derating with power setpoint $P_\mathrm{set}$ (see steady state calculation in Appendix B).

To sum up, the MRS-based derating significantly decreases the thrust force for $\Omega = 1$, i.e. if no further rotor acceleration is

455    feasible. This is in line with the observed reduction of damage equivalent loads during derating, as reported by Aho et al. (2014).

Otherwise, for $\Omega < 1$, i.e. especially at low wind speeds or minor derating in region II, the MRS-based derating accelerates the rotor, but the resulting increase in thrust force is negligible.

**4.2 WT inertia provision for the reference frequency event**

This section evaluates WT grid-forming capabilities in terms of maximum inertia provision as a function of WT operating point $(v_{\mathrm{w}}, P_{\mathrm{set}})$. At first, Sect. 4.2.1 introduces the considered reference frequency event defined by the German grid code and derives a worst-case test scenario for WT inertia provision. Then, Sect. 4.2.2 discusses the resulting dynamic WT simulations for optimized inertia provision, i.e. for $H_{\mathrm{v}} = H_{\mathrm{v,max}}$, with and without MRS-based derating. Finally, Sect. 4.2.3 discusses the mapping of WT operating points to the maximum feasible inertia constant over a wide operating range.

**4.2.1 Grid codes**

Although the grid codes can vary between countries and system operators, the core requirements for inertia provision are similar (Ghimire et al., 2024). This paper focuses on the German code for grid-forming control and their requirements for inertia provision (VDE, 2024a, b). This grid code defines two reference frequency events with maximum initial ROCOF magnitudes of $|\dot{f}_{\mathrm{s}}| = 2\,\mathrm{Hz\,s^{-1}}$: one for negative inertia provision due to a high positive initial ROCOF, and another for positive inertia provision due to a high negative initial ROCOF . The latter is considered as the worst-case reference frequency event for WFs, since the output power has to increase for inertia provision, which decelerates the WTs. Emulating this reference frequency event and evaluating the WF power response is required to verify inertia provision.

The considered grid code (VDE, 2024b) defines various tests based on the reference frequency events to verify inertia provision, including operation in (i) fictive or simulated island mode with changing power imbalance $\Delta P$ due to varying electrical load, (ii) grid-emulator-connected mode with changing ROCOF $\dot{\Omega}_{\mathrm{s}}$, and (iii) real grid-connected mode with changing controller-internal ROCOF, corresponding to the VSM acceleration $\dot{\Omega}_{\mathrm{v}}$ (see also Fig. 5). In the latter case (iii), the frequency signal defined by the reference frequency events is added as a disturbance to the controller-internal frequency, corresponding to the VSM speed $\Omega_{\mathrm{v}}$. It should be noted that some tests consider deactivated droop control. However, the verification principle is always the same, see also Schöll et al. (2024). In all tests, a ROCOF $\dot{\Omega}$ changes and the inertial power response $\Delta P$ is measured, or vice versa. The actual inertia provision is quantified by the measured inertia constant $H_{\mathrm{meas}} = |\Delta P/(2\dot{\Omega})|$ at quasi-steady state, see also Eq. (1a). Note that $\Delta P$ corresponds to the measured power change $\Delta P_{\mathrm{meas}}$ only if droop control is deactivated. Otherwise, the droop power change $\Delta \overline{P}_{\mathrm{d}}$ (depending on the frequency deviation) adds to the inertial power (depending on the ROCOF) during the frequency event, i.e. $\Delta P = \Delta P_{\mathrm{meas}} - \Delta \overline{P}_{\mathrm{d}}$ for correct $H_{\mathrm{meas}}$-calculation. Clearly, $H_{\mathrm{meas}}$ must match the expected VSM inertia constant, i.e. $H_{\mathrm{meas}} \approx H_{\mathrm{v}}$.

For the considered VSM control, $H_{\mathrm{meas}} \approx H_{\mathrm{v}}$ holds if no power saturation is active (see also Fig. 6). In other words, it is assumed that the actual VSM control implementation would pass all tests (VDE, 2024b) with $H_{\mathrm{meas}} \approx H_{\mathrm{v}}$ for an arbitrarily chosen $H_{\mathrm{v}}$ if no power saturation exists. It follows that $H_{\mathrm{meas}} \approx H_{\mathrm{v}}$ holds if no operating limits are violated in the simulations of the WT model (see Fig. 2). Otherwise, in reality, the protection methods would saturate the output power as the desired

inertia provision is not feasible, i.e. $H_{\mathrm{meas}} < H_{\mathrm{v}}$ due to $H_{\mathrm{v}} > H_{\mathrm{v,max}}$. Thus, assuming proper VSM control allows this study to focus on the relevance of interactions with WT control, WT characteristics, and operating limits.

490    Running and passing all tests defined in the grid code (VDE, 2024b) verifies proper grid-forming control implementation and inertia provision for a given $H_{\mathrm{v}}$ at some given operating points. However, this approach is not suitable for evaluating the maximum feasible inertia provision of WFs over a wide operating range. Thus, this paper considers a single worst-case test scenario to simulate WTs for varying $H_{\mathrm{v}}$ and varying operating point $(v_{\mathrm{w}}, P_{\mathrm{set}})$, see the corresponding inputs in Fig. 2. The reference frequency event for positive inertia provision with an initial ROCOF of $\dot{f}_{\mathrm{s}} = -2\,\mathrm{Hz\,s^{-1}}$ (VDE, 2024b) defines

495    the input $\dot{\Omega}_{\mathrm{s}}$ in Fig. 2. This can be interpreted as ideal ROCOF emulation at the point of common coupling. As in real grid-connected operation mode, droop control is activated. Finally, a WT survives the worst-case test scenario if no operating limit is violated, i.e. for $H_{\mathrm{v}} \le H_{\mathrm{v,max}}$ at a given operating point $(v_{\mathrm{w}}, P_{\mathrm{set}})$ according to Eq. (3).

**4.2.2 Optimized time response**

This section considers simulation results of the WT inertial response to the reference frequency event with optimized $H_{\mathrm{v}} =$

500    $H_{\mathrm{v,max}}$ at $v_{\mathrm{w}} = 9\,\mathrm{m\,s^{-1}} < v_{\mathrm{w,R}}$ for two different power setpoints (i) $P_{\mathrm{set}} = 100\,\%$ for MPPT in Fig. 9 and (ii) $P_{\mathrm{set}} = 95\,\%$ for MRS-based derating in Fig. 10. The grid frequency is identical in both cases (panel a), i.e. $\Omega_{\mathrm{s}}$ decreases with the initial ROCOF $\dot{\Omega}_{\mathrm{s}} = -4\,\%\,\mathrm{s^{-1}}$ for $t_0 = 0\,\mathrm{s} \le t < 1\,\mathrm{s}$ and further decreases with $\dot{\Omega}_{\mathrm{s}} = -2/3\,\%\,\mathrm{s^{-1}}$ afterwards, until reaching the nadir $\Omega_{\mathrm{s,min}} = 95\,\%$ at $t_{\mathrm{s,min}} = 2.5\,\mathrm{s}$. Then, $\Omega_{\mathrm{s}}$ increases with $\dot{\Omega}_{\mathrm{s}} = 2\,\%\,\mathrm{s^{-1}}$, before staying constant at $\Omega_{\mathrm{s}} = 98\,\%$ for $t \ge 4\,\mathrm{s}$. In addition, panel (a) shows the VSM speed $\Omega_{\mathrm{v}}$, the WT speed $\Omega$, and the adjusted MPPT input $\Omega_{H0}$. Panel (b) shows the

505    mechanical power $P_{\mathrm{m}}$, the electrical power $P_{\mathrm{e}}$, the electrical torque $M_{\mathrm{e}}$ and its limit $M_{\mathrm{e,max}}$. Panel (d), (e), and (f) show the tip speed ratio $\lambda$, the power coefficient $c_{\mathrm{p}}$, and the blade pitch angle $\beta$, respectively. Finally, panel (c) shows the resulting aerodynamic trajectory (red line).

In Fig. 9, the power coefficient starts at its maximum value due to MPPT, i.e. $(\beta, \lambda) = (\beta^\star, \lambda^\star)$ which yields $c_{\mathrm{p}} = c_{\mathrm{p}}^\star$. The VSM inertial response increases the electrical power $P_{\mathrm{e}}$ during the negative ROCOF, whereas the aerodynamic or mechanical

510    power $P_{\mathrm{m}}$ remains almost constant. Thus, the WT rotor decelerates such that the tip speed ratio $\lambda$ decreases. After the ROCOF changes from negative to positive at $t_{\mathrm{s,min}} = 2.5\,\mathrm{s}$, $P_{\mathrm{e}}$ rapidly decreases below $P_{\mathrm{m}}$ such that the rotor accelerates. Accordingly, considering the trajectory $(c_{\mathrm{p}}, \lambda)$ in Fig. 9c, the WT leaves its MPP $(\lambda^\star, c_{\mathrm{p}}^\star)$ during the negative ROCOF, with significantly decreasing $\lambda$ but almost constant $c_{\mathrm{p}}$. Due to the grid synchronization delay of the VSM, the WT reaches its rotor speed nadir or $(\lambda_{\mathrm{min}}, c_{\mathrm{p,min}})$ at $t = 2.593\,\mathrm{s}$, i.e. shortly after the frequency nadir at $t_{\mathrm{s,min}} = 2.5\,\mathrm{s}$. Afterwards, with increasing $\Omega$, the trajectory

515    converges to $(\lambda^\star, c_{\mathrm{p}}^\star)$ for $t \to \infty$ again.

At $t \approx 4.65\,\mathrm{s}$ in Fig. 9a, the MPPT compensation resets $\Omega_{H0}$ to $\Omega$ such that $P_{\mathrm{e}}$ in panel (b) rapidly decreases to re-accelerate the WT to its MPP, called WT rotor speed recovery. Clearly, decreasing the rate limit $\dot{\Omega}_{H0,\mathrm{max}}$ for the transition between active and inactive MPPT compensation in Eq. (20) leads to a smoother change of $P_{\mathrm{e}}$, which is expected to cause less severe secondary frequency disturbances (Godin et al., 2019). However, this would slow down the WT rotor speed recovery. Finding

520    a reasonable compromise is beyond the scope of this paper, but see Appendix G for further discussion.

[Figure]

**Figure 9.** WT simulation results for MPPT at $\left(v_{\mathrm{w}} = 9\,\mathrm{m\,s^{-1}}, P_{\mathrm{set}} = 100\,\%\right)$ with optimized $H_{\mathrm{v,max}} = 2.26\,\mathrm{s}$.

[Figure]

**Figure 10.** WT simulation results for MRS-based derating at $\left(v_{\mathrm{w}} = 9\,\mathrm{m\,s^{-1}}, P_{\mathrm{set}} = 95\,\%\right)$ with optimized $H_{\mathrm{v,max}} = 3.80\,\mathrm{s}$.

In Fig. 10, the initial steady-state power $P_\mathrm{m}(t_0) = P_\mathrm{e}(t_0)$ is $5\,\%$ below its initial MPP value in Fig. 9 due to $P_\mathrm{set} = 95\,\%$ or $c_\mathrm{p}(t_0) = 95\,\% \cdot c_\mathrm{p}^\star$ with $t_0 = 0$. Note that the lower initial electromagnetic torque $M_\mathrm{e}(t_0)$ leads to (i) higher initial WT speed $\Omega(t_0) = 1$ in Fig. 10 compared with $\Omega(t_0) = 91.8\,\%$ in Fig. 9, and (ii) active pitch control, i. e. $\beta(t_0) > \beta^\star$ in Fig. 10. The trajectory $(c_\mathrm{p}, \lambda)$ in Fig. 10c starts at $(\lambda_0, c_\mathrm{p,0})$ with $\lambda_0 > \lambda^\star$ and $c_\mathrm{p,0} < c_\mathrm{p}^\star$. The VSM inertial response increases the electromagnetic power $P_\mathrm{e}$ during the negative ROCOF such that the WT decelerates and $\lambda$ decreases. At the same time, the pitch control decreases $\beta$ due to $\Omega < 1$. The trajectory reaches the (local) WT rotor speed nadir or $(\lambda_1, c_\mathrm{p,1})$ at $t \approx 2.62\,\mathrm{s}$. Afterwards, during the positive ROCOF, $P_\mathrm{e}$ rapidly decreases such that $\lambda$ increases, reaching $(\lambda_2, c_\mathrm{p,2})$ at $t \approx 4.44\,\mathrm{s}$. Finally, during constant but below-rated frequency $\Omega_\mathrm{s} < 1$ for $t \geq 4$, inertia provision is inactive, but the active power droop control increases $P_\mathrm{e}$ to the maximum available MPPT power, which is greater than the initial value $P_\mathrm{e}(t_0)$. Thus, $(c_\mathrm{p}, \lambda)$ converges to the MPP $(\lambda^\star, c_\mathrm{p}^\star)$ for $t \to \infty$. Note that the trajectory in Fig. 10 converges to the MPP from the right side of the MPP, whereas, for $P_\mathrm{set} = 100\,\%$ in Fig. 9, the trajectory converges to the MPP from the left side of the MPP.

After the ROCOF changes from negative to positive at $t = 2.5\,\mathrm{s}$ in Fig. 10d, the electrical power $P_\mathrm{e} \approx P_\mathrm{ref} = \overline{P}_\mathrm{set}^\star + P_\mathrm{v} + \overline{P}_\mathrm{d}$ (panel b) decreases as the VSM inertial power becomes negative, i. e. $P_\mathrm{v} < 0$. At the same time, the droop power increases as power reserves become available, i. e. $\overline{P}_\mathrm{d} = P_\mathrm{d,max} > 0$ (see also Fig. 2). The droop power $\overline{P}_\mathrm{d} > 0$ counteracts the VSM inertial power $P_\mathrm{v} < 0$ during the grid frequency recovery. Thus, both droop control and inertia provision are active at the minimum $P_\mathrm{e}$ at $t = 4\,\mathrm{s}$ in Fig. 10b, whereas $\overline{P}_\mathrm{d} = P_\mathrm{d,max} = 0$ and $P_\mathrm{v} > 0$ holds for the maximum $P_\mathrm{e}$ at $t = 1\,\mathrm{s}$.

Considering Figs. 9 – 10 (panel b), the electromagnetic torque $M_\mathrm{e}$ reaches its limit $M_\mathrm{e,max}$ in both cases, i. e. the second constraint of the optimization problem Eq. (3) is active. However, the initial torque $M_\mathrm{e}(t_0)$ for $5\,\%$ power derating in Fig. 10b is more than $12\,\%$ lower than $M_\mathrm{e}(t_0)$ for MPPT in Fig. 9b. The torque reduction is higher than the power derating due to an $11\,\%$ higher initial rotor speed $\Omega(t_0)$ in Fig. 10a than in Fig. 9a. Clearly, the MRS-based derating maximizes the torque headroom $M_\mathrm{e,max} - M_\mathrm{e}(t_0)$ for inertia provision by maximizing the rotor speed.

In general, the VSM inertial response to the initial negative ROCOF increases the electrical power $P_\mathrm{e}$, which decelerates the rotor in both cases (Figs. 9 – 10). However, due to a higher initial rotor speed, the rotor speed does not fall below its MPP-value $\Omega = 91.84\,\%$ for MRS-based derating in Fig. 10; whereas $\Omega$ decreases to $88.15\,\%$ for MPPT in Fig. 9. Moreover, the WT rotor deceleration leads to constant (or only slightly decreasing) mechanical power $P_\mathrm{m}$ in Fig. 9 but increasing $P_\mathrm{m}$ in Fig. 10. Consequently, the MRS-based derating strategy provides both additional kinetic energy and additional wind energy reserves for inertia provision.

To summarize, the optimized values for the VSM inertia constant $H_\mathrm{v} = H_\mathrm{v,max}$ are $2.26\,\mathrm{s}$ and $3.80\,\mathrm{s}$ for $P_\mathrm{set} = 100\,\%$ (MPPT) in Fig. 9 and for $P_\mathrm{set} = 95\,\%$ (MRS-based derating) in Fig. 10, respectively. For these two exemplary operating points, the MRS-based derating of $5\,\%$ increases the inertia provision capability in terms of $H_\mathrm{v,max}$ by ca. $3.80/2.26 - 1 \approx 68\,\%$ compared with MPPT.

**4.2.3 Mapping of WT operating points**

The proposed WF inertia forecasting approach uses LUTs to map the operating points of all WTs to the WF inertia (see also Fig. 1), assuming optimal tuning $H_\mathrm{v} = H_\mathrm{v,max}$ for each WT in the WF. For the proposed solution of the optimization problem

in Eq. (3), Fig. 11 depicts the resulting LUTs of the form $H_{\mathrm{v,max}} = f(v_{\mathrm{w}}, P_{\mathrm{set}})$ for all operating points $(v_{\mathrm{w}}, P_{\mathrm{set}})$ within the range $v_{\mathrm{w,cut\text{-}in}} \leq v_{\mathrm{w}} \leq 15\,\mathrm{m\,s}^{-1}$ and $90\,\% \leq P_{\mathrm{set}} \leq 100\,\%$. For enhanced visualization and clarity, the panels (a) and (b) show the same results from different perspectives. In the 2-D plot (panel a), the color coding for the power setpoint $P_{\mathrm{set}}$ is the same as in Fig. 8; whereas, in the 3-D plot (panel b), the datapoint colors indicate the active optimization constraints in Eq. (3).

[Figure]

**Figure 11.** Maximum inertia provision at different operating points $(v_{\mathrm{w}}, P_{\mathrm{set}})$, with active constraints highlighted in panel (b).

In Fig. 11, the rotor speed limit $\Omega_{\mathrm{min}}$ is only active for low wind speeds near $v_{\mathrm{w,cut\text{-}in}}$. For most operating points with $v_{\mathrm{w}} < 6.3\,\mathrm{m\,s}^{-1}$, the torque rate limit $\dot{M}_{\mathrm{e,max}}$ constraint is active. For $P_{\mathrm{set}} = 1$, i. e. even without derating, the maximum inertia constant $H_{\mathrm{v,max}} = 4.55\,\mathrm{s}$ at $v_{\mathrm{w}} = 6.3\,\mathrm{m\,s}^{-1}$ exceeds the rated WT physical inertia constant $H_{\mathrm{R}} = 3.26\,\mathrm{s}$. However, for $6.3\,\mathrm{m\,s}^{-1} < v_{\mathrm{w}} \leq v_{\mathrm{w,R}}$, the torque limit $M_{\mathrm{e,max}}$ reduces $H_{\mathrm{v,max}}$ with increasing wind speed. For above-rated wind speeds $v_{\mathrm{w}} > v_{\mathrm{w,R}}$ and $P_{\mathrm{set}} = 1$, the virtual inertia constant $H_{\mathrm{v,max}}$ is smaller than the WT physical inertia constant $H = H_{\mathrm{R}}$ due to the electrical power limit $P_{\mathrm{e,max}}$. The MRS-based derating increases $H_{\mathrm{v,max}}$ over the complete wind speed range. [5]

For each operating point $(v_{\mathrm{w}}, P_{\mathrm{set}})$ and corresponding optimal $H_{\mathrm{v}} = H_{\mathrm{v,max}}$ tuning, the rotor speed nadir $\min \Omega$ and the mechanical power extrema $\min P_{\mathrm{m}}$ or $\max P_{\mathrm{m}}$ are evaluated in the inertial response time interval defined in the appendix Eq. (J1). Figure 12a shows the normalized deviation of $\min \Omega$ to the initial WT speed $\Omega_0$, i. e. $(\min \Omega - \Omega_0)/\Omega_0$, and Fig. 12b shows the normalized deviation of $\min \Omega$ to the MPP speed $\Omega_{\mathrm{mpp}}$, i. e. $(\min \Omega - \Omega_{\mathrm{mpp}})/\Omega_{\mathrm{mpp}}$, where $\Omega_{\mathrm{mpp}} := \Omega_0$ for $P_{\mathrm{set}} = 1$. Figures 13a and 13b show the normalized deviation of $\min P_{\mathrm{m}}$ and $\max P_{\mathrm{m}}$ to the initial WT mechanical power $P_{\mathrm{m,0}}$, i. e. $(\min P_{\mathrm{m}} - P_{\mathrm{m,0}})/P_{\mathrm{m,0}}$ and $(\max P_{\mathrm{m}} - P_{\mathrm{m,0}})/P_{\mathrm{m,0}}$, respectively.

The WT significantly decelerates during the inertial response at low wind speeds, although the rotor speed deviation does not exceed $20\,\%$ of $\Omega_0$ in Fig. 12a. Clearly, the lower speed limit $\Omega_{\mathrm{min}}$ is only relevant for a few operating points at low wind speeds and minor derating, i. e., for all other operating points the available kinetic energy reserve cannot be fully extracted for inertia provision due to active torque or power constraints. Decreasing $P_{\mathrm{set}}$ increases $\Omega_0$ such that the rotor speed does not fall below its MPP value if the derating is high enough, i. e. for $\min \Omega - \Omega_{\mathrm{mpp}} > 0$ in Fig. 12b. In this case, no rotor speed recovery is needed after the inertial response.
* * *
[5]See Appendix G for $H_{\mathrm{v,max}}$-results with an additional (optional) constraint for WT rotor speed recovery and their discussion. See Appendix H for the $H_{\mathrm{v,max}}$-results of the simplified solution derived in Sect. 3.3.2 and a comparison with the complete numerical solution.

For MPPT, the rotor deceleration decreases the power coefficient $c_p$ (see also Fig. 3). Thus, the aerodynamic power decreases during the inertial response, i. e. $\min P_m < P_{m,0}$ for $P_{set} = 1$ at below-rated wind speeds in Fig. 13a. However, minor derating significantly increases $\min P_m$. For $P_{set} \leq 95\,\%$, the reduction of $P_m$ is negligible due to $\Omega_0 \approx \Omega_{mpp}$, or $P_m$ even increases

580 due to $\min \Omega > \Omega_{mpp}$, see Figs. 12a and 13a. In summary, the MRS-based derating leads to less severe rotor deceleration due to higher initial rotor speed and thus also higher aerodynamic power during the inertial response.

[Figure]

**Figure 12.** WT rotor speed nadir: deviation from the initial operating point **(a)** and the MPP **(b)**.

[Figure]

**Figure 13.** WT mechanical power extrema: minimum **(a)** and maximum **(b)** deviation from the initial operating point.

**4.3 WF inertia monitoring and forecasting**

Applying the proposed approach to a real WF ambient wind profile over five days, the following figures illustrate:

1. actual and forecasted WF ambient wind condition inputs for the wake model (Fig. 14);

585 2. WF inertia monitoring results, i. e. WF inertia calculations based on actual wind conditions (Figs. 15 – 16);

3. WF inertia forecasting results, i. e. WF inertia calculations based on predicted wind conditions (Fig. 17);

4. uncertainties due to WF ambient wind forecast errors, wake model errors, and WT inertia provision errors (Figs. 17 – 19).

[Figure]

**Figure 14.** Ambient wind conditions and its hour-ahead forecasts: wind speed **(a)** and wind direction **(b)** for actual measurements (Actual) as well as for deterministic ("Det"), probabilistic minimal ("Pro Min"), and probabilistic maximal ("Pro Max") forecasts.

[Figure]

**Figure 15.** Simulation results for all 12 WTs operating at $P_{set} = 1$ (MPPT) based on the actual WF ambient wind data in Fig. 14.

In addition to the actual WF ambient wind speed and direction, Fig. 14a shows the different hour-ahead forecasts. The wind speed $v_w$ values of the deterministic ("Det") forecast lie between the values of the probabilistic minimal ("Pro Min") forecast

590 and the probabilistic maximal ("Pro Max") forecast. The forecast for the wind direction $\Gamma_{\mathrm{w}}$ in Fig. 14b is deterministic. All
forecast trends match the actual data. The following WF simulation results are based on actual and forecasted WF ambient
wind data for inertia monitoring and forecasting, respectively.

In Fig. 15a, the maximum (local) wind speed (upper envelope curve) of all twelve WTs (WT1 – WT12) equals the actual
WF ambient wind speed in Fig. 14. Clearly, wake effects reduce the wind speed for downstream WTs, resulting in a lower
595 mean wind speed in Fig. 15. In Fig. 15b, the normalized WT power at steady state is defined as $P := P_{\mathrm{e}} = P_{\mathrm{m}}$. Due to the
equal power rating of all WTs, the normalized WF power corresponds to the mean normalized WT power, and the WF (virtual)
inertia constant corresponds to the mean WT (virtual) inertia constant. For $P_{\mathrm{set}} = 1$, the WF operates at rated WF power $P = 1$
if the local wind speed at all WTs reaches $v_{\mathrm{w}} \geq v_{\mathrm{w,R}}$, e. g. at $t = 110\,\mathrm{h}$ in Fig. 15b. Accordingly, at $t = 110\,\mathrm{h}$ in Fig. 15c, all
WTs operate at $\Omega = 1$ such that the physical inertia constant $H$ is saturated by its rated value $H_{\mathrm{R}}$. In contrast, the maximum
600 virtual inertia constant $H_{\mathrm{v,max}}$ in Fig. 15d is saturated by its lower limit given by the maximum power constraint, see also
Fig. 11. For lower wind speeds with all WTs operating in region II, e. g. at $t = 70\,\mathrm{h}$ in Fig. 15, the lower WT rotor speeds lead
to lower $H$ but higher $H_{\mathrm{v,max}}$, as the power constraint becomes inactive. Note that $H_{\mathrm{v,max}} > H$ is possible because the WTs
rotate asynchronously, whereas the VSMs synchronize with the grid frequency. Thus, for a given ROCOF, a VSM-controlled
WT with physical inertia constant $H$ and virtual inertia constant $H_{\mathrm{v}} > H$ may extract more kinetic energy than a (directly
605 grid-connected) SM with the same $H$.

[Figure]

**Figure 16.** Simulation results at WF level for different derating power setpoints $P_{\mathrm{set}}$ based on the actual WF ambient wind data in Fig. 14.

In Fig. 16, assuming equal $P_{\text{set}}$ for all WTs, simulation results are considered for different $P_{\text{set}}$ at the WF level. The WF available power $\overline{P}^\star$ increases with lower $P_{\text{set}}$ (see panel a), i.e. $\overline{P}^\star - P_{\text{mppt}} > 0$ for $P_{\text{set}} < 1$ holds most of the time (see panel c), where $P_{\text{mppt}} := P = \overline{P}^\star$ for $P_{\text{set}} = 1$. This is due to wake effects, i.e. derating increases the local wind speeds at the downstream WTs. In Fig. 16b, the actual WF power $P$ decreases with lower $P_{\text{set}}$, but minor changes are visible for lower wind speeds, (i) the derating power setpoint is normalized to the available (and not to the rated) power, i.e. $P_{\text{set}} = P/\overline{P}^\star$, and since (ii) higher $\overline{P}^\star$ for lower $P_{\text{set}}$ partially compensates for derating according to $P = P_{\text{set}}\overline{P}^\star$, see also Eq. (16). Thus, the actual WF derating fraction is defined as $P_{\text{rel}} := P/P_{\text{mppt}}$ in Fig. 16d. $P_{\text{rel}}$ is higher than $P_{\text{set}} = P/\overline{P}^\star$ (dashed lines) most of the time, as derating results in higher available power $\overline{P}^\star$. In Fig. 16e, derating increases $H$ according to the MRS up to the limit $H = H_{\text{R}}$. Besides additional kinetic energy reserve, the MRS-based derating provides wind energy reserve and power headroom for inertia provision. Thus, in Fig. 16f, $H_{\text{v,max}}$ increases with lower $P_{\text{set}}$, even for $H = H_{\text{R}}$.

[Figure]

**Figure 17.** Comparison of actual and forecasted WF inertia, with errors (forecasted minus actual values) of the forecast types: deterministic ("Det"), probabilistic minimal ("Pro Min"), probabilistic maximal ("Pro Max"), and combined minimal ("Comb Min").

Figure 17 shows the actual and forecasted $H_{\text{v,max}}$-values at WF level for exemplary power setpoints $P_{\text{set}} = 1$ (panel a) and $P_{\text{set}} = 96\,\%$ (panel b), with the different wind input data for the wake model given by Fig. 14. The additional combined minimum ("Comb Min") forecast in Fig. 17 uses the "Pro Min" and "Pro Max" forecasts. More precisely, the "Comb Min" forecast finds the minimum $H_{\text{v,max}}$-values within the forecasted local wind speed range $v_{\text{w,pro,min}} \leq v_{\text{w}} \leq v_{\text{w,pro,max}}$ for each WT, where $v_{\text{w,pro,min}}$ and $v_{\text{w,pro,max}}$ are given by wake modeling with "Pro Min" and "Pro Max" forecast input data, respectively. Since $H_{\text{v,max}}$ is a nonlinear function of $v_{\text{w}}$ for a given $P_{\text{set}}$, a nonlinear optimization algorithm solves $\min H_{\text{v,max}}(v_{\text{w}})$ such

that $v_{\mathrm{w,pro,min}} \leq v_{\mathrm{w}} \leq v_{\mathrm{w,pro,max}}$ for each WT. Again, the considered WF values are given by the mean values across all WTs. In Figures 17a and 17b, the "Comb Min" and "Pro Max" forecasts are similar most of the time, but notable deviations occur especially during time intervals with lower wind speeds, which is in line with Fig. 11.

625 Figures 17c and 17d show the "Det" and "Comb Min" inertia forecast errors for different derating. Due to lower $H_{\mathrm{v,max}}$ variation for lower derating (see panels a, b), the absolute errors tend to be smaller in these cases. During almost all hours, the "Comb Min" forecast error is negative, i.e. the "Comb Min" forecast predicts a lower bound for $H_{\mathrm{v,max}}$. This lower bound is especially relevant for (i) WF operators if they have to provide a minimum level of inertia, or for (ii) system operators to analyze worst-case frequency events.

[Figure]

**Figure 18.** Inertia monitoring errors of the simplified optimization ("Simple Opt") variant and the no wake modeling ("No Wake") variant. For each considered variant and for each $P_{\mathrm{set}} \in \{0.9, 0.92, 0.94, 0.96, 0.98, 1\}$, all 120 WF ambient wind data samples (one data point per hour) of the actual measurements in Fig.14 are mapped to $H_{\mathrm{v,max}}$ and to the corresponding errors (variant minus actual/proposed). The bars in panels c and f include all $6 \cdot 120 = 720$ samples per variant, with negative mean value ("−MEAN") equal to the mean absolute error (MAE), sample standard deviation ("STD"), and root mean square error (RMSE).

630 Figure 18 compares inertia monitoring errors due to simplified WF modeling for two variants: (i) replacing the numerical by the simplified solution of the optimization problem in Eq. (3) leads to the "Simple Opt" variant, see also Sect. 3.3.2; (ii) no wake modeling leads to the "No Wake" variant. Figures 18a–c consider the absolute error ("Abs. Error") $\Delta H_{\mathrm{v,max}}$ (in s) and Figures 18c–f consider the relative error ("Rel. Error") $\Delta H_{\mathrm{v,max}}/H_{\mathrm{v,max}}$ (in %). Both variants underestimate the WF inertia

at all operating points, i.e. $\Delta H_{v,max} = -|\Delta H_{v,max}|$, such that the negative mean value ("−MEAN") and the mean absolute error (MAE) are equal. The MAEs are $0.37\,$s and $1.22\,$s (panel c) or $11.3\,\%$ and $38.5\,\%$ (panel f) for the "Simple Opt" and "No Wake" variants, respectively. The error magnitude tends to increase with higher derating for the "Simple Opt" variant (see panels a, d), which is in line with the analysis at the WT level in Appendix H.

[Figure]

**Figure 19.** Inertia forecasting errors of the proposed ("Proposed") and simplified variants ("Simple Opt", "No Wake"). For each considered variant and for each $P_{set} \in \{0.9, 0.92, 0.94, 0.96, 0.98, 1\}$, all 120 WF ambient wind data samples (one data point per hour) of the deterministic ("Det") forecast in Fig.14 are mapped to $H_{v,max}$. Thus, the shown distributions include $6 \cdot 120 = 720$ samples per variant, with negative mean value ("−MEAN"), sample standard deviation ("STD"), mean absolute error (MAE), and root mean square error (RMSE).

In Figures 19a and 19c, the WF inertia forecast error distributions of the "Simple Opt" and "No Wake" variants are shifted to the left compared with the proposed variant due to the inertia underestimation in Fig. 18. At the samples with $\Delta H_{v,max} > 0$ in Fig. 19a, the wind forecast errors overcompensate for the modeling errors $\Delta H_{v,max} < 0$ in Fig. 18a. For the proposed complete numerical WF inertia forecasting, the MAE is $0.9\,$s or $26.8\,\%$, see Figs. 19b and 19d. For the "Simple Opt" forecasting, the MAE is $1.04\,$s or $30.3\,\%$. For the "No Wake" forecasting, the MAE is $1.68\,$s or $46.5\,\%$. Thus, the simplified variants "Simple Opt" and "No Wake" increase the WF inertia forecasting MAE (in s) by $\frac{1.04}{0.9} - 1 \approx 16\,\%$ and $\frac{1.68}{0.9} - 1 \approx 87\,\%$, respectively.

**5    Conclusions**

645    Grid-forming VSM control limits the initial ROCOF by inertia provision and thus enables grid frequency stability in future power systems with a high share of IBRs. VSM inertia can be adapted based on the required grid support. VSM control extracts WT kinetic energy reserve for inertia provision, but physical and virtual WT inertia differ in general. The WT with physical inertia constant $H$ rotates asynchronously, whereas the VSM with virtual inertia constant $H_{\mathrm{v}}$ synchronizes with the grid frequency. However, in contrast to the grid-following control without inertia provision, the rotor speed and grid frequency

650    dynamics are not completely decoupled anymore, i. e. the VSM requires the WT energy or power for grid synchronization. Thus, WT operating limits must be taken into account for proper grid synchronization with unsaturated inertia provision. Furthermore, the consideration of intra-farm turbine-to-turbine interactions is of utmost significance, as WFs will participate in future inertia markets. This could be additionally facilitated by short-term prediction of the maximum feasible inertia capability of the WF over varying inflow and operational conditions.

655    In contrast to existing solutions, the proposed formulation considers the WF grid-forming capability in terms of the maximum feasible inertia constant $H_{\mathrm{v,max}}$ by simulating the inertial response of VSM-controlled WTs. Furthermore, the proposed formulation takes into account the intra-farm turbine-to-turbine interactions, as they have a huge influence on the local wind condition at the turbines, and have largely been ignored in the existing studies evaluating inertia provision capability. Under varying wind conditions, the derived simplified solution without dynamic WT simulations exhibits a trend similar to that of the

660    complete numerical solution with dynamic WT simulations. However, the simplified solution significantly underestimates the optimal tuning value $H_{\mathrm{v}} = H_{\mathrm{v,max}}$ for maximum inertia provision. In addition, the proposed dynamic WT simulations give deeper insights into the WT inertial response, including the relevance of operating limits and the interactions among different controllers, such as VSM control, MPPT compensation, MRS-based derating, active power droop control, and blade pitch control.

665    The proposed MRS-based derating increases the WT kinetic energy reserve for inertia provision with negligible impact on the thrust force. However, upon reaching the maximum rotor speed, further derating increases the pitch angle, significantly reducing thrust force and wake effects. Regarding the aerodynamic trajectory of the power coefficient $c_{\mathrm{p}}(t)$ as a function of the tip ratio $\lambda(t)$ during the inertial response to the worst-case ROCOF, the MRS-based derating provides kinetic energy and power headroom for inertia provision. This is achieved by initial operation with $(\lambda(t_0) > \lambda^{\star}, c_{\mathrm{p}}(t_0) < c_{\mathrm{p}}^{\star})$ at the right side of

670    the MPP $(\lambda^{\star}, c_{\mathrm{p}}^{\star})$, i. e. the power coefficient $c_{\mathrm{p}}(t)$ increases during WT deceleration. If the initial derating and thus the initial rotor speed $\Omega_0$ is high enough, the rotor speed $\Omega(t)$ remains above its MPP value during the inertial response. In this case, no rotor speed recovery is required afterwards. The simulation results show that the proposed MRS-based derating increases the WF grid-forming capability in terms of $H_{\mathrm{v,max}}$ over the complete wind speed range. For instance, considering a below-rated wind speed of $v_{\mathrm{w}} = 9\,\mathrm{m\,s}^{-1}$ in Sect. 4.2.2, a derating of $5\,\%$ significantly increases $H_{\mathrm{v,max}}$ by ca. $68\,\%$.

675    WT curtailment or derating increases the energy or power reserve for inertia provision, but wasting renewable energy should be avoided. Therefore, quantifying and forecasting inertia is essential for maximizing efficiency while ensuring grid stability. This paper demonstrated a generic approach for quantifying and forecasting the inertia provision of a WF based on weather

forecasting, wake modeling, and mapping local WT operating points to grid-forming capabilities. The actual WF power loss is smaller than expected by local WT derating setpoints, since derating reduces wake effects and, thus, increases the available power. The $H_{v,max}$ forecast error depends on the uncertainty of the WF ambient wind forecast. Taking this into account, the proposed lower bound prediction for $H_{v,max}$ combines probabilistic minimal and maximal wind forecasts. Even for this conservative estimation, $H_{v,max}$ varies over time, and the WF can provide more inertia than at rated power during many hours. Besides the proposed wind forecasting, the WF inertia forecasting includes (i) the proposed wake modeling and (ii) the proposed optimization of the WT inertial response to avoid high errors due to (over)simplification.

Future work should validate the simulated WT inertial response based on high-fidelity modeling, e. g. including multi-mass modeling or based on real measurements. However, the modeling of the proposed generic approach can be readily adjusted, if necessary. Utilizing the presented simulation results for WF control is straightforward when operating all WTs with the same power setpoint $P_{set}$ for derating. However, future work may also consider an optimal WF control that distributes individual $P_{set}$-values to each WT, e. g. to maximize the WF power while providing the desired inertia. Based on the new grid codes and inertia market requirements, the future mechanical and electrical designs of WTs and WFs should consider enhanced capabilities for grid-forming control, including inertia provision.

**Appendix A:  Simplified torque rate constraint**

Limiting the torque rate $\dot{M}_e$ in Eq. (3) contradicts the simplifying assumption of an ideal power pulse in Eq. (6) with infinite torque rate. However, instead of neglecting the actual limit $\dot{M}_{e,max}$, a simplified torque rate constraint can be derived as follows. Transforming Eq. (18) in the time domain, the VSM torque response to a ROCOF step change from zero to $\dot{\Omega}_s$ is given by

$$M_{e,v}(t) = \dot{\Omega}_s \frac{4H_v + D_v t}{2} e^{-\frac{D_v}{4H_v}t} - 2H_v \dot{\Omega}_s. \tag{A1}$$

By differentiating with respect to time one gets

$$\dot{M}_{e,v}(t) = -\dot{\Omega}_s \frac{D_v{}^2 t}{8H_v} e^{-\frac{D_v}{4H_v}t}, \tag{A2}$$

and

$$\ddot{M}_{e,v}(t) = -\dot{\Omega}_s \frac{D_v{}^2}{8H_v} \left(1 - \frac{D_v}{4H_v}t\right) e^{-\frac{D_v}{4H_v}t}. \tag{A3}$$

Zeroing Eq. (A3) and inserting the solution into Eq. (A2) yields the maximum VSM torque rate

$$\max \dot{M}_{e,v}(t) = \dot{M}_{e,v}\left(t = \frac{4H_v}{D_v}\right) = -\frac{1}{2e}\dot{\Omega}_s D_v. \tag{A4}$$

Neglecting the WT deceleration until reaching the maximum torque rate, i. e. $\Omega(t) \approx \Omega_0$, and scaling Eq. (A4) by $1/\Omega_0$ due to asynchronous WT rotation with respect to VSM speed or grid frequency, the simplified WT torque rate constraint is given by

$$\max \dot{M}_e = \frac{\max \dot{M}_{e,v}}{\Omega_0} = \frac{\dot{\Omega}_{s,max}}{2e\Omega_0} D_v(H_v) \leq \dot{M}_{e,max}. \tag{A5}$$

**Appendix B: Steady state calculation**

Solving the optimization problem in Eq. (3) for a given WT operating point defined by $(v_\mathrm{w}, P_\mathrm{set})$ requires initializing the simulation model to a steady state. However, the steady-state conditions are unknown in general. Instead of running time-consuming simulations until reaching steady state, an alternative is proposed to speed up the optimization. At the equilibrium point, mechanical and electrical power are equal, i.e. the initial steady states $(\Omega_0, \beta_0)$ solve the minimization problem

$$(\Omega_0, \beta_0) := \arg\min |e(\Omega, \beta)| \tag{B1}$$

where $\quad e(\Omega, \beta) := P_\mathrm{m} - P_\mathrm{e} = P_\mathrm{w}(v_\mathrm{w}) c_\mathrm{p}(\Omega, \beta, v_\mathrm{w}) - \Omega M_\mathrm{e}(\Omega, v_\mathrm{w}, P_\mathrm{set}) = 0.$

The following analysis evaluates $(v_\mathrm{w}, P_\mathrm{set})$ and identifies active constraints in the different WT operating regions to convert Eq. (B1) into simpler one-dimensional subproblems.

Firstly, for MPPT with $P_\mathrm{set} = 1$ and $M_\mathrm{e} = M_\mathrm{mppt}$, Eq. (B1) can be rewritten as (Thommessen and Hackl, 2024)

$$(\overline{\Omega}^\star, \overline{\beta}^\star) := \arg\min |\overline{e}^\star(\Omega, \beta)| \qquad \text{s.t.} \begin{cases} \Omega_\mathrm{min} \leq \Omega \leq 1, \\ \beta_\mathrm{min} \leq \beta \leq \beta_\mathrm{max}, \\ \beta_\mathrm{min} = \beta & \text{if } v_\mathrm{w} \leq v_\mathrm{w,R}, \\ 1 = \Omega & \text{if } v_\mathrm{w} > v_\mathrm{w,R}, \end{cases} \tag{B2}$$

where $\quad \overline{e}^\star(\Omega, \beta) := P_\mathrm{m} - \Omega M_\mathrm{e} = P_\mathrm{w} c_\mathrm{p} - \Omega M_\mathrm{mppt}$, for a given wind speed $v_\mathrm{w}$.

With minimum region-II wind speed $v_\mathrm{w,II,min}$, one-dimensional optimizations solve

$$\begin{cases} v_\mathrm{w,cut\text{-}in} \leq v_\mathrm{w} < v_\mathrm{w,II,min} \Rightarrow \begin{pmatrix} \overline{\Omega}^\star \\ \overline{\beta}^\star \end{pmatrix} = \begin{pmatrix} \arg\min |\overline{e}^\star| \\ \beta_\mathrm{min} = \beta_\mathrm{min}(\lambda) \end{pmatrix} \\ v_\mathrm{w,II,min} \leq v_\mathrm{w} \leq v_\mathrm{w,R} \Rightarrow \begin{pmatrix} \overline{\Omega}^\star \\ \overline{\beta}^\star \end{pmatrix} = \begin{pmatrix} \lambda^\star \frac{v_\mathrm{w}}{\omega_\mathrm{R} r} \\ \beta_\mathrm{min} = \beta^\star \end{pmatrix} \\ v_\mathrm{w,R} < v_\mathrm{w} \leq v_\mathrm{w,cut\text{-}out} \Rightarrow \begin{pmatrix} \overline{\Omega}^\star \\ \overline{\beta}^\star \end{pmatrix} = \begin{pmatrix} 1 \\ \arg\min |\overline{e}^\star| \end{pmatrix} \end{cases} \tag{B3}$$

with either $\overline{\Omega}^\star$ or $\overline{\beta}^\star$ known in advance (except for the transition region I-II in the first case, where $\overline{\beta}^\star = \beta_\mathrm{min}(\lambda)$ is calculated based on $\lambda = \overline{\Omega}^\star \omega_\mathrm{R} r / v_\mathrm{w}$). For the available WT power $\overline{P}^\star$ in Eq. (15), the MPPT power coefficient is given by

$$\overline{c}_\mathrm{p}^\star(v_\mathrm{w}) = \begin{cases} c_\mathrm{p}(\overline{\Omega}^\star, \overline{\beta}^\star, v_\mathrm{w}) & \text{if } v_\mathrm{w,cut\text{-}in} \leq v_\mathrm{w} < v_\mathrm{w,II,min}, \\ c_\mathrm{p}^\star := \max c_\mathrm{p} & \text{if } v_\mathrm{w,II,min} \leq v_\mathrm{w} \leq v_\mathrm{w,R}, \\ \frac{1}{P_\mathrm{w}(v_\mathrm{w})} & \text{if } v_\mathrm{w,R} < v_\mathrm{w} \leq v_\mathrm{w,cut\text{-}out}, \end{cases} \tag{B4}$$

Secondly, for derating with $P_\mathrm{set} < 1$ and $P_\mathrm{e} = \overline{P}^\star P_\mathrm{set}$, Eq. (B1) can be rewritten as

$$(\Omega_0, \beta_0) := \arg\min |\widetilde{e}(\Omega, \beta)| \tag{B5}$$

where $\quad \widetilde{e}(\Omega, \beta) := P_\mathrm{m} - P_\mathrm{e} = p_\mathrm{w} c_\mathrm{p} - \overline{P}^\star P_\mathrm{set}$, which is solved in two substeps.

In the first substep, ignoring the speed limit by assuming inactive pitch control, leads to the theoretical steady-state rotor speed

$$\widetilde{\Omega}_0 := \arg\min |\widetilde{e}(\Omega, \beta = \beta_{\min})| \tag{B6}$$

with $\beta_{\min} = \beta_{\min}(\lambda)$. The second substep takes the speed limit $\Omega = 1$ into account, i. e.

$$\quad \Rightarrow \begin{cases} \widetilde{\Omega}_0 \leq 1 \Rightarrow \begin{pmatrix} \Omega_0 \\ \beta_0 \end{pmatrix} = \begin{pmatrix} \widetilde{\Omega}_0 \\ \beta_{\min} \end{pmatrix} \\ \widetilde{\Omega}_0 > 1 \Rightarrow \begin{pmatrix} \Omega_0 \\ \beta_0 \end{pmatrix} = \begin{pmatrix} 1 \\ \arg\min |\widetilde{e}| \end{pmatrix}. \end{cases} \tag{B7}$$

This way, the conversion of the original two-dimensional optimization problem in Eq. (B1) into simpler one-dimensional subproblems as in Eq. (B3), (B6), (B7) has been achieved.

**Appendix C: Grid impedance and electromagnetic feedback for VSM control**

The grid impedance comprises several physical impedances, such as transformer or line impedances, but also include virtual
735 impedances emulated by inverter control (Taul et al., 2020; VDE, 2024a). Besides, for type 3 WTs, which use DFIMs, the physical grid impedance includes the DFIM stator impedance (Thommessen and Hackl, 2024). For type 4 WTs, which use full-scale back-to-back inverters, the physical grid impedance includes LC(L)filter impedances connected to the grid-side inverter (Taul et al., 2020). A high grid impedance (corresponding to a weak grid connection) leads to a lower electromagnetic feedback gain $k_e$ (Thommessen and Hackl, 2024, Sect. III.E) and a lower natural angular velocity $\omega_{n,v}$ in Eq. (18), resulting in
740 a lower inertial response time. In contrast, a higher grid voltage increases $k_e$ and $\omega_{n,v}$ (VDE, 2024a).

From the perspective of system operators or grid stability, minimum $k_e$ or $\omega_{n,v}$ values characterize the worst case in terms of a weak grid connection and a slow inertial response. For instance, the weak grid connection of (offshore) WFs is characterized by a typical short circuit ratio SCR $< 2$ (Ghimire et al., 2024). Thus, Ghimire et al. (2024) recommend an operating point sweep between maximum and minimum SCR for testing grid-forming capabilities and (load angle) stability analysis.
745 From the perspective of WF operators or WT protection, the worst case is characterized by maximum $k_e$ or $\omega_{n,v}$ values due to fast inertial response with high torque rates. Accordingly, the considered worst-case analysis of the WT grid-forming capability should assume maximum $k_e$ or $\omega_{n,v}$ values. Grid codes requirements (VDE, 2024a) implicitly define an admissible $k_e$ or $\omega_{n,v}$ value range by specifying lower and upper limits for the grid impedance and voltage.

**Appendix D: VSM torque versus power synchronization**

750 Replacing the torque with power quantities in Fig. 5 results in a power instead of a torque synchronization loop (Roscoe et al., 2020). However, the normalized VSM power and torque are approximately equal (see Eq. 19). For type 4 WTs with full-scale back-to-back inverters, the VSM-controlled grid-side inverter (approximately) outputs the VSM power (see Appendix E),

which is in line with the proposed simplified representation. However, for type 3 WTs with DFIMs, the VSM control is not implemented at the grid-side but at the machine-side inverter connected to the DFIM rotor (see Appendix F). Considering VSM torque synchronization, the DFIM emulates the VSM torque rather than the VSM power (see Appendix F). In this case, the WT inertial *power* response (quasi-steady-state value) depends on the DFIM or WT rotor speed $\Omega$ according to $P_e = \Omega M_e$. Consequently, for type 3 WTs with VSM torque synchronization, it would be necessary to distinguish between (i) the VSM parameter $H_v$, representing the inertial *torque* response $\Delta M_e = M_{e,v} \approx P_v \approx -2H_v\dot{\Omega}_s$ (see also Eq. 19) and (ii) a power-equivalent inertia constant $H_{eq}$ for inertia provision, representing the inertial *power* response, e. g. $\Delta P_e \approx \Delta M_e \Omega =: -2H_{eq}\dot{\Omega}_s$, with $\Omega \neq \Omega_s \approx 1$ in general. In contrast, for type 4 WTs, $H_v$ defines the inertial *power* response of both WTs and VSMs, i. e. $\Delta P_e = P_v \approx -2H_v\dot{\Omega}_s$, and no additional definition of $H_{eq}$ is needed for quantifying inertia provision.

For simplicity, this paper considers type 4 WTs, for which torque and power synchronization loops result in similar inertial responses. For type 3 WTs with *torque* synchronization, the proposed generic approach is also applicable by taking $H_{eq} \neq H_v$ into account. In this case, the VSM representation in Fig. 2 must simply be adjusted by redefining the VSM power in Eq. (17) as $P_v := \Omega M_{e,v}$. For type 3 WTs with *power* synchronization, it must be simply taken into account that the VSM synchronization speed $\omega_{n,v}$ depends on the DFIM speed (see Appendix F).

**Appendix E: VSM control for type 4 WTs and assumptions**

For type 4 WTs based on full-scale back-to-back inverters, the grid-side and machine-side inverters are connected via a DC-link. The grid-side inverter tracks the WT power reference based on VSM control, whereas the machine-side inverter controls the DC-link voltage $u_{dc}$ (Roscoe et al., 2020; Meseguer Urban et al., 2019; Nguyen et al., 2022). Strictly speaking, the power of the machine-side and grid-side inverter are *not* equal during $u_{dc}$ transients due to the DC-link buffer energy $E_{dc} = \frac{1}{2}C_{dc}u_{dc}^2$ with DC-link capacity $C_{dc}$. However, $E_{dc}$ is usually negligible in comparison with the WT kinetic energy reserve, i. e. $E_{dc} \ll E_{kin}$, see also Hackl et al. (2018). Thus, $E_{dc}$ is usually *not* relevant for inertia provision. Moreover, a small $E_{dc}$ requires a fast-reacting or aggressively tuned DC-link voltage control to keep $u_{dc}$ within its small admissible voltage range (Thommessen and Hackl, 2024). Thus, this paper neglects the power difference or delay between machine-side and grid-side inverters.

**Appendix F: VSM control for type 3 WTs and assumptions**

For type 3 WT based on DFIMs, the directly grid-connected DFIM stator generates most of the power. The DFIM rotor is connected to the grid via back-to-back inverters, with the power flow depending on the DFIM slip $\Omega - \Omega_s$ (Dirscherl and Hackl, 2016). With the VSM torque defined as the DFIM torque, i. e. $M_{e,v} := M_e$, the DFIM stator power approximately equals the VSM power, i. e. $P_{stator} = \Omega_s M_e \approx P_v$ (Rodriguez-Amenedo et al., 2021; Thommessen and Hackl, 2024). However, the total electrical DFIM power includes the DFIM rotor power as well, i. e. $P_e = P_{stator} + P_{rotor} = \Omega_s M_e + (\Omega - \Omega_s) M_e = \Omega M_e \neq P_v$ (Dirscherl and Hackl, 2016). Alternatively, Shah and Gevorgian (2020) propose grid-forming control for DFIMs based on a power instead of a torque synchronization loop with a power feedback $P_v := P_e$ instead of a torque feedback $M_{e,v} := M_e$. In

this case, the DFIM electrical power feedback gain $k_e$ varies with the WT rotor speed $\Omega$ according to $P_e = \Omega M_e$, and thus the VSM inertial response time or $\omega_{n,v}$ in Eq. (18) also varies.

**Appendix G:  WT rotor speed recovery and requirements**

Assuming initial operation at the maximum power point (MPP), the WT deceleration during the inertial response decreases the aerodynamic power (see Fig. 9). Thus, during the so-called recovery phase after the frequency event, the output power has to decrease below the initial power to maintain speed. In fact, the WT power is reduced even further to (slowly) re-accelerate the rotor, which affects the grid frequency recovery and may lead to a secondary frequency drop (Bao et al., 2016; Duckwitz, 2019; Höhn et al., 2024). Thus, a former draft of the German grid code (VDE, 2024a, Version 0.1) explicitly prohibited reducing the output power for speed recovery. More precisely, after the ROCOF returns to zero at $t = 4\,\mathrm{s}$ in Fig. 9, reducing the output power below its initial value to re-accelerate the rotor would not be allowed. However, this constraint is probably more restrictive than necessary, as discussed next.

In fact, the frequency event defined in the grid code (VDE, 2024b) may not represent a realistic situation, but can be regarded as a worst-case test to verify the inertia provision capability of a grid-connected unit. Godin et al. (2019) consider a frequency event with less severe ROCOFs, which requires a lower power increase $\Delta P$ over a longer period $\Delta t$, before reaching the frequency nadir. However, the kinetic energy extraction $\Delta P \Delta t$ may be similar to the one used in our work. Although Godin et al. (2019) assume a slower frequency recovery afterwards, the rotor speed recovery is similar, since speed recovery takes much longer than frequency recovery (see Fig. 9). Considering the power reduction due to MPP deviations during the recovery phase of a real WF, Godin et al. (2019) observe that similar reductions can be expected simply from changes in wind conditions. Moreover, the inertial response superimposes all variations in WT operating points, which, in reality, results in a smoothing effect on the grid frequency during the recovery phase (see also Thommessen and Hackl, 2024). Consequently, the requirements for the recovery power should not be too restrictive. As per the recent version of the German grid code (VDE, 2024a), no recovery power constraints are considered in this paper by default, although they can be easily included if necessary, as demonstrated next.

[Figure]

**Figure G1.** Proposed solution with additional recovery power constraint: maximum virtual inertia constant for different operating points $(v_w, P_{set})$ and with active optimization constraints highlighted in panel (b).

The optimization results in Fig. G1 include an additional constraint for the recovery phase after the WT inertial response. More precisely, the power $P_{\text{mppt}}$, available for rotor speed recovery after the frequency event ends at $t_{\text{end}} := 4.1\,\text{s}$, is not allowed to fall more than $1\,\%$ below the initial power, i. e. $P_{\text{rec,min}} := P_{\text{m,0}} - 1\,\% \leq P_{\text{mppt}}(\Omega(t_{\text{end}}))$. This $P_{\text{rec,min}}$ constraint significantly limits the inertia provision at below-rated wind speeds for low derating, since most of the WT kinetic energy reserve cannot be used. In this case, the WT rotor speed limit $\Omega_{\text{min}}$ is irrelevant, as the $P_{\text{rec,min}}$ constraint is active instead.

**Appendix H: Results of the simplified solution for maximum inertia provision**

The results of the simplified solution derived in Sect. 3.3.2 are shown in the left plot of Fig. H1. In comparison with the results of the complete numerical solution in Fig. 11, the simplified solution underestimates the WT inertia provision capability. Considering the right plot of Fig. H1, significant simplification errors can be observed, e. g. $|\Delta H_{\text{v,max}}| > 1\,\text{s}$ at $v_{\text{w}} = 7\,\text{m s}^{-1}$ for $P_{\text{set}} \leq 96\,\%$. For $P_{\text{set}} = 1$, one reason for the underestimation is that the simplified solution assumes a constant electrical power in Eq. (6), whereas in reality, the VSM inertial power $P_{\text{v}}$ decreases with $\Omega_{\text{v}} \approx \Omega_{\text{s}}$ in Eq. (17). The underestimation significantly increases with higher derating, since the simplified solution only considers kinetic energy reserve and does not account for wind power reserve or changing aerodynamics. More precisely, the MRS-based derating leads to initial operation at the right side of the MPP. During the inertial response, the decreasing $\lambda$ in combination with decreasing $\beta$ increases the aerodynamic or mechanical power, which counteracts the WT rotor deceleration, as also shown in Fig. 10.

[Figure]

**Figure H1.** Simplified solution: maximum virtual inertia constant for different operating points **(a)** and simplification error or deviation from the complete numerical solution **(b)**.

**Appendix I: Prioritization of inertia provision over droop control**

Both the inertial response and droop control are essential for grid frequency stability. Firstly, the inertial response is always needed to limit the magnitude of ROCOF regardless of its sign, i.e., even during frequency recovery. Otherwise, if the ROCOF is too high, grid-connected resources trip, e.g., due to loss of synchronism or load angle instability. Secondly, active power

droop control (or primary frequency control) is needed to limit the steady-state frequency deviation. Although WTs can reduce their power to support over-frequency events, WTs in MPPT mode cannot permanently increase their power for prolonged support of under-frequency events. Thus, droop control only has to increase the WT output power only if quasi-steady power reserves are available due to previous derating (VDE, 2024a). A temporary droop control activation, i.e. a temporary power offset proportional to the frequency deviation based on kinetic energy extraction, is often proposed for a fast frequency response of grid-following WTs (Godin et al., 2019; Lee et al., 2016). Although this would temporarily increase frequency support, the slower frequency response would hide the urgency of power adaptation for grid support from other grid-connected resources. More precisely, the slower frequency change or lower frequency deviation would delay or decrease the power response from other resources. This may also lead to a secondary frequency event after the temporary support. Thus, such a temporary droop control activation is not considered in the German grid code for grid-forming technologies (VDE, 2024a) or in this work.

In summary, the inertial response limits the ROCOF, which determines the time available to react, e.g. for protection devices to shed loads, or for other resources to ramp up power production. Thus, the inertial response is of highest priority and uses all power reserves, including kinetic energy. With increasing frequency deviation, droop control (or primary frequency control) limits steady-state frequency deviation. For under-frequency events, droop control is only feasible for WTs if wind power reserves are available due to previous derating. WTs should *not* use kinetic energy reserves for temporary droop control activation in order to properly communicate the droop power demand to other resources via the grid frequency.

**Appendix J: Further definitions and parameters**

The inertial response time interval for evaluating extreme WT operating points in Figs. 12 – 13 is defined as

$$
t_0 \leq t \leq \min\{t_1, t_2\}, \quad t_1 := \max\left\{t_{s,min}, \arg\min \Omega(t)\right\}, \quad t_2 := \left\{ \begin{array}{l} t_1, \text{ if } \forall t \geq t_0 : \overline{P}_d(t) \leq 0, \\ \min t \text{ s. t. } \overline{P}_d(t) > 0 \wedge t \geq t_{s,min}, \text{ otherwise,} \end{array} \right\} \quad \text{(J1)}
$$

where $t_1$ takes into account that the WT speed nadir may occur (shortly) after the grid frequency nadir at $t_{s,min}$, and $t_2$ takes into account that the droop control may cause further uncritical WT deceleration for $t \geq t_{s,min}$.

**Appendix K: Nomenclature**

| | |
|---|---|
| DFIM | Doubly-fed induction machine |
| FCNN | Fully connected neural networks |
| HVDC | High voltage DC current |
| IBR | Inverter-based resources |
| LUT | Lookup tables |
| MAE | Mean absolute error |
| MPP | Maximum power point |
| MPPT | Maximum power point tracking |

| | | |
|---|---|---|
| | MRS | Maximum rotation strategy |
| | NWP | Numerical weather prediction |
| | OP | Operating point |
| | PGPE | Policy gradients with parameter-based exploration |
| 860 | RMSE | Root mean square error |
| | ROCOF | Rate of change of frequency |
| | SCR | Short circuit ratio |
| | SM | Synchronous machines |
| | STD | Standard deviation |
| 865 | VSM | Virtual synchronous machine |
| | WF | Wind farm |
| | WT | Wind turbine |

| | | |
|---|---|---|
| 870 | $c_\mathrm{p}$ | Power coefficient |
| | $c_\mathrm{t}$ | Thrust coefficient |
| | C | Capacitance |
| | $d_\mathrm{t}$ | Damping coefficient of tower spring-mass-damper system |
| | D | Diameter |
| 875 | $D$ | Damping factor |
| | $\Delta P$ | Power imbalance |
| | $\Delta t$ | Time duration |
| | $E_\mathrm{kin}$ | Kinetic energy |
| | $f$ | Frequency |
| 880 | $F_\mathrm{t}$ | Aerodynamic thrust force |
| | $F_\mathrm{w}$ | Force of the incoming wind |
| | $H$ | Inertia constant |
| | $k_\mathrm{e}$ | Electromagnetic feedback gain |
| | $k_\mathrm{t}$ | Stiffness coefficient of tower spring-mass-damper system |
| 885 | $m$ | Torque |
| | $M$ | Non-dimensional torque |
| | $m_\mathrm{t}$ | Mass of tower spring-mass-damper system |
| | $n$ | Total number of turbines |
| | $\overline{c}_\mathrm{p}$ | Power coefficient corresponding to the available power |
| 890 | $\overline{P}$ | Non-dimensional available power |

| | | |
|---|---|---|
| $p$ | | Power |
| $P$ | | Non-dimensional power |
| $P_{\mathrm{w}}$ | | Power of the incoming wind |
| $r$ | | Radius of the rotor |
| $s$ | | Laplace-domain variable |
| $s_{\mathrm{t}}$ | | Tower-top fore-aft displacement |
| $t$ | | Time variable |
| $u$ | | Voltage |
| $v_{\mathrm{w}}^{u}$ | | North-aligned component of the wind measurement |
| $v_{\mathrm{w}}^{v}$ | | East-aligned component of the wind measurement |
| $v_{\mathrm{w}}$ | | Wind speed |
| $\widetilde{v}_{\mathrm{w}}$ | | Relative wind speed |
| $w$ | | Weight variable |
| $x$ | | State variable |

| | | |
|---|---|---|
| $\beta$ | | Blade pitch angle |
| $\delta$ | | Load angle |
| $\epsilon$ | | Threshold |
| $\lambda$ | | Tip-speed ratio |
| $\mu$ | | Mean |
| $\omega$ | | Angular frequency |
| $\omega_{\mathrm{n}}$ | | Natural angular frequency of a second order system |
| $\Omega$ | | Non-dimensional angular frequency |
| $\phi$ | | Electrical angle |
| $\rho$ | | Air density |
| $\sigma$ | | Standard deviation |
| $\Theta$ | | Moment of inertia |
| $\zeta$ | | Damping ratio of a second order system |

| | | |
|---|---|---|
| $\square_{\mathrm{comb}}$ | | Value that combines min. and max. probabilistic forecasts |
| $\square_{\mathrm{d}}$ | | Droop |
| $\square_{\mathrm{det}}$ | | Deterministic forecast value |
| $\square_{\mathrm{dp}}$ | | Damping |

895

900

905

910

915

920

925

| | | |
|---|---|---|
| | $\square_e$ | Electrical |
| | $\square_m$ | Mechanical |
| | $\square_{max}$ | Maximum |
| | $\square_{meas}$ | Measured |
| 930 | $\square_{min}$ | Minimum |
| | $\square_{mpp}$ | Value at the maximum power point |
| | $\square_{mppt}$ | Value for maximum power point tracking |
| | $\square_R$ | Rated |
| | $\square_{ref}$ | Reference |
| 935 | $\square_{pu}$ | Per unit value |
| | $\square_{rec}$ | Value for an additional WT rotor speed recovery constraint |
| | $\square_s$ | System / Grid |
| | $\square_{set}$ | Setpoint |
| | $\square_{pro}$ | Probabilistic forecast value |
| 940 | $\square_v$ | Virtual / VSM |
| | $\square_0$ | Initial / steady-state value |

| | | |
|---|---|---|
| | $\square^{[i]}$ | Value for $i$-th turbine |
| 945 | $\square^*$ | Optimal value |
| | $\dot{\square}$ | Time derivative $\mathrm{d}\square/\mathrm{d}t$ |

*Data availability.* The content and data of figures 3-4, 6-18, G1, and H1 can be retrieved in Python pickle format via the DOI[6]: https://doi.org/10.5281/zenodo.15176373.

950 *Author contributions.* AT, AA and CMH developed the formulation of WF inertia forecasting. AT and AA carried out the research, with CLB developing the concept of including wake effects. AT and CMH evaluated the grid-forming capability of WTs, whereas AT implemented WT modeling and control with inputs from AA and CMH. AA developed the ambient wind forecaster based on the formulation proposed by CLB, developed and implemented the operational constraints with CLB, and implemented the wind farm model with inputs from AT. AT prepared the manuscript, with contributions from AA, CMH and CLB particularly in the sections about forecasting and farm modelling. AT
955 generated and interpreted the inertia forecasting results based on the actual and forecasted data that AA provided. CLB and CMH supervised the overall research. All authors provided important input to this research work through discussions, feedback, and improved this manuscript.
* * *
[6]Review note: This is a preliminary repository, the final DOI will be generated if the paper is accepted.

*Competing interests.* The authors declare that they have no conflict of interest, except for CLB who is the Editor-in-Chief of the Wind Energy Science journal.

*Financial support.* This work has been supported by the SUDOCO and TWAIN projects, which receive funding from the European Union's Horizon Europe Programme under the grant agreements No. 101122256 and 101122194, respectively. This work has also been partially supported by the e-TWINS project (FKZ: 03EI6020), which received funding from the German Federal Ministry for Economic Affairs and Climate Action (BMWK).

*Acknowledgements.* The authors express their gratitude to Mr. Benjamin Dittrich and Mr. David Coimbra from EnergieKontor AG, who granted access to the field data.

960